# Timely vaccine strain selection and genomic surveillance improve evolutionary forecast accuracy of seasonal influenza A/H3N2

John Huddleston[1]*, Trevor Bedford[1,2]

[1]Vaccine and Infectious Disease Division, Fred Hutchinson Cancer Center, Seattle, United States; [2]Howard Hughes Medical Institute, Seattle, United States

## eLife Assessment

This study investigates the influence of genomic information and timing of vaccine strain selection on the accuracy of influenza A/H3N2 forecasting. The authors utilized appropriate statistical methods and have provided **convincing** evidence, which amounts to an **important** contribution to the evidence base. Substantial revisions have been made to the manuscript and issues of concern have been clarified, with the necessary study limitations appropriately discussed.

*For correspondence:
jhuddles@fredhutch.org

Competing interest: The authors declare that no competing interests exist.

**Abstract** Evolutionary forecasting models inform seasonal influenza vaccine design by predicting which current genetic variants will dominate in the influenza season 12 months later. Forecasting models depend on hemagglutinin sequences from global public health networks to identify current genetic variants (clades) and estimate clade fitnesses. The lag between collection of a clinical sample and public availability of its sequence averages ~3 months, complicating the 12-month forecasting problem by reducing our understanding of current clade frequencies. Despite continued methodological improvements to forecasting models, these constraints of a 12-month forecast horizon and 3-month submission lags impose an upper bound on any model's accuracy. The SARS-CoV-2 pandemic revealed that modern vaccine technology reduces forecast horizons to 6 months and expanded sequencing support reduces submission lags to 1 month on average. We quantified the potential effects of these public health policy changes on forecast accuracy for A/H3N2 populations. Reducing forecast horizons to 6 months reduced average absolute forecasting errors to 25% of the 12-month average, while reducing submission lags decreased uncertainty in current clade frequencies by 50%. These results show the potential to improve the accuracy of existing forecasting models through realistic changes to public health policy.

## Introduction

Seasonal influenza virus infections cause approximately half a million deaths per year (*World Health Organization, 2014*). Vaccination provides the best protection against hospitalization and death, but the rapid evolution of the influenza surface protein hemagglutinin (HA) allows viruses to escape existing immunity and requires regular updates to influenza vaccines (*Petrova and Russell, 2018*). The World Health Organization (WHO) meets twice a year to decide on vaccine updates for the Northern and Southern Hemispheres (*Morris et al., 2018*). The dominant influenza vaccine platform is an inactivated whole virus vaccine grown in chicken eggs (*Wong and Webby, 2013*), which takes 6–8 months to develop, contains a single representative vaccine virus per seasonal influenza subtype including A/

H1N1pdm, A/H3N2, and B/Victoria (*Morris et al., 2018*), and for which only the HA protein content is standardized (*Yamayoshi and Kawaoka, 2019*). These constraints require the WHO to select a single virus per subtype that is immunologically representative of the next season's dominant HA approximately 12 months before the peak of that next season. These selections depend on the diversity of currently circulating phylogenetic clades, groups of influenza viruses that all share a recent common ancestor. The WHO's understanding of that genetic diversity comes from HA sequences collected by the WHO's Global Influenza and Surveillance and Response System (*Hay and McCauley, 2018*) and submitted to the Global Initiative on Sharing All Influenza Data (GISAID) EpiFlu database (*Shu and McCauley, 2017*). The fastest evolving influenza subtype A/H3N2 accumulates 3–4 HA amino acid (AA) substitutions per year (*Smith et al., 2004*; *Kistler and Bedford, 2023*) such that the clades circulating 12 months after the vaccine decision can be antigenically distinct from clades that were circulating at the time of the decision.

Given the 12-month lag between the decision to update an influenza vaccine and the peak of the following influenza season, the vaccine composition decision is commonly framed as a long-term forecasting problem (*Lässig et al., 2017*). For this reason, the decision process is partially informed by computational models that attempt to predict the genetic composition of seasonal influenza populations 12 months in the future (*Morris et al., 2018*). The earliest of these models predicted future influenza populations from HA sequences alone (*Luksza and Lässig, 2014*; *Neher et al., 2014*; *Steinbrück et al., 2014*). Recent models include phenotypic data from serological experiments (*Morris et al., 2018*; *Huddleston et al., 2020*; *Meijers et al., 2023*; *Meijers et al., 2025*). Since most serological experiments occur after genetic sequencing (*Hampson et al., 2017*) and all forecasting models depend on HA sequences to determine the viruses circulating at the time of a forecast, sequence availability is the initial limiting factor for any influenza forecasts. Unfortunately, the average lag between collection of a seasonal influenza A/H3N2 HA sample and submission of its sequence had been ~3 months in the era prior to the SARS-CoV-2 pandemic (*Figure 1A*). While long-term forecasting models continue to improve technically, the constraints of a 12-month forecast horizon and the availability of enough recent, representative HA sequences impose an upper bound on the accuracy of long-term forecasts.

The global response to the SARS-CoV-2 pandemic in 2020 showed the speed with which we can develop new vaccines and capture real-time viral genetic diversity. Decades of research on mRNA vaccines enabled the development of multiple effective vaccines a year after the emergence of SARS-CoV-2 (*Mulligan et al., 2020*; *Baden et al., 2021*). This mRNA-based vaccine platform also enabled the approval of booster vaccines targeting Omicron only 3 months after the recommendation of an Omicron-based vaccine candidate (*Grant et al., 2023*). In parallel to vaccine development, expanded funding and capacity building for viral genome sequencing enabled unprecedented dense sampling of a pathogen's genetic diversity over a short period of time (*Chen et al., 2022*). By 2021, the average time between collection of a SARS-CoV-2 sample and submission of the sample's genome sequence to GISAID EpiCoV database had decreased to approximately 1 month (*Brito et al., 2022*). This reduction in submission lags reflects both increased emergency funding and the sustained efforts by more public health organizations to adopt best practices for genomic epidemiology (*Kalia et al., 2021*; *Black et al., 2020*). Assessments of SARS-CoV-2 short-term forecasts have shown how such reductions in forecast horizon and submission lags can improve the accuracy of short-term forecasts and real-time estimates of clade frequencies (*Abousamra et al., 2024*).

These technological and public health policy changes in response to SARS-CoV-2 suggest that we could realistically expect the same outcomes for seasonal influenza. Work on mRNA vaccines for influenza viruses dates back over a decade (*Petsch et al., 2012*; *Brazzoli et al., 2016*; *Pardi et al., 2018*; *Feldman et al., 2019*), and multiple vaccines have completed phase 3 trials by early 2025 (*Soens et al., 2025*; *Pfizer, 2022*). A switch from the current egg-based inactivated virus vaccines to mRNA vaccines could reduce the time between vaccine design decisions and the peak influenza season from 12 months to 6 months. Similarly, the expanded global capacity for sequencing SARS-CoV-2 genomes could reasonably extend to broader and more rapid genomic surveillance for seasonal influenza, reducing submission lags from 3 months to 1 month on average. Even in the years immediately after the onset of the SARS-CoV-2 pandemic, we have observed a trend toward a reduced average submission lag of 2.5 months that we would expect from increased global capacity for genome sequencing (*Figure 1—figure supplement 1*).

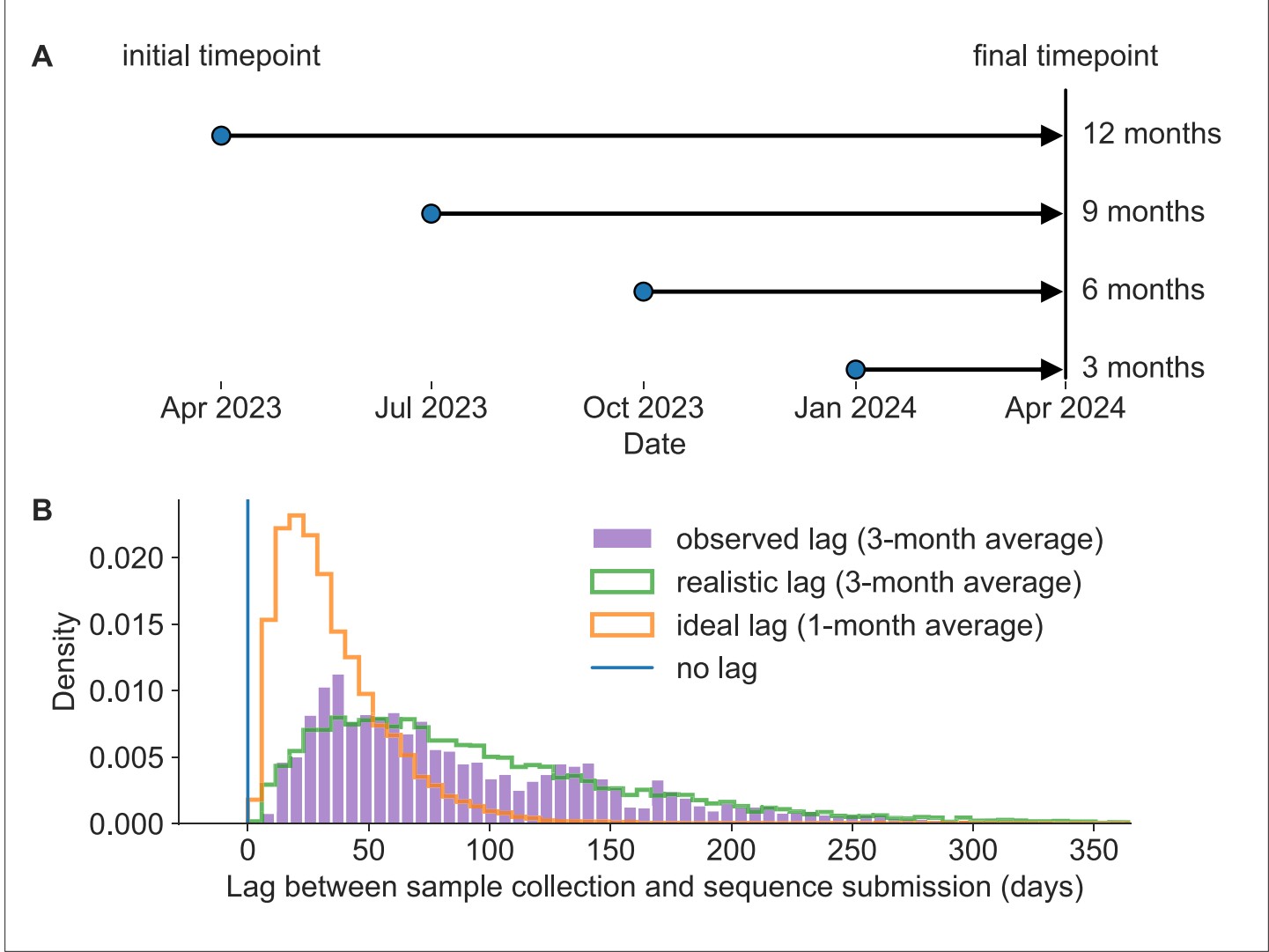

**Figure 1.** Model of forecast horizons and submission lags. (**A**) Long-term forecasting models historically predicted 12 months into the future from April and October because of the time required to develop and distribute a new vaccine (*Luksza and Lässig, 2014*). We tested three additional shorter forecast horizons in 3-month intervals of 9, 6, and 3 months prior to the same time in the future season. For each forecast horizon, we calculated the accuracy of forecasts under each of the three submission lags described below including no lag, realistic lag, and ideal lag. (**B**) Observed lags in days between collection of viral samples and submission of corresponding hemagglutinin (HA) sequences to Global Initiative on Sharing All Influenza Data (GISAID) (purple) for samples collected in 2019 have a mean of 98 days (approximately 3 months). A gamma distribution fit to the observed lag distribution with a similar mean and shape (green) represents a realistic submission lag that we sampled from to assign "submission dates" to simulated and natural A/H3N2 populations. A gamma distribution with a mean that is one-third of the realistic distribution (orange) represents an ideal submission lag analogous to the 1-month average observed lags for SARS-CoV-2 genomes. Retrospective analyses including fitting of forecasting models typically filter HA sequences by collection date instead of submission dates in which case there is no lag (blue).

The online version of this article includes the following source data and figure supplement(s) for figure 1:

**Source data 1.** Distribution of lags between sample collection and sequence submission in prepandemic and pandemic eras; see distribution_of_submission_lags.csv at https://doi.org/10.5281/zenodo.17259448.

**Figure supplement 1.** Distribution of submission lags in days for the pre-pandemic era (2019–2020) and pandemic era (2022–2023 in orange).

**Figure supplement 2.** Number and proportion of A/H3N2 sequences available per timepoint and lag type.

**Figure supplement 3.** Number and proportion of simulated A/H3N2-like sequences available per timepoint and lag type.

**Figure supplement 4.** Number of all available sequences per region and year and proportion of sequences sampled by two different subsampling methods.

In this work, we tested the effects of similar reductions in forecast horizons and submission lags on the accuracy of long-term forecasts for seasonal influenza. Building on our previously published forecasting framework (*Huddleston et al., 2020*), we performed a retrospective analysis of HA sequences from simulated and natural A/H3N2 populations. For each population type, we produced forecasts from 12, 9, 6, and 3 months prior to a given influenza season (*Figure 1A*). We made each forecast under three different submission lag scenarios, including a realistic lag (3 months on average), an ideal lag (1 month on average), and no lag (*Figure 1B*). First, we measured the accuracy and precision of forecasts under these different scenarios by calculating the genetic distance between predicted and observed future populations using the same earth mover's distance metric that we originally used to train our forecasting models (*Rubner et al., 1998*). Next, we calculated the effect of forecast horizon and submission lags on clade frequencies which are the values we use to communicate predictions to WHO decision-makers (*Huddleston et al., 2024*). We quantified the effect of reduced submission lags on initial clade frequencies, and we calculated forecast accuracy as the difference between predicted and observed clade frequencies of future populations. Finally, we calculated the relative improvement in forecast accuracy produced by different realistic interventions including reduced vaccine development time, reduced submission lags, and the combination of both. In this way, we show the potential to improve the accuracy of existing long-term forecasting models and, thereby, the quality of vaccine design decisions by simplifying the forecasting problem through realistic societal changes.

## Results

### Reducing forecast horizons and submission lags decreases distances between predicted and observed future populations

Previously, we trained long-term forecasting models that minimized the genetic distance between predicted and observed future populations of HA sequences (*Huddleston et al., 2020*). We predicted each population 12 months in the future based on the frequencies and fitness estimates of HA sequences in the current population. We calculated the distance between predicted and observed future populations with the earth mover's distance metric (*Rubner et al., 1998*). This metric provided an average genetic distance between AA sequences of the two populations weighted by the frequencies of sequences in each population. This approach allowed us to measure forecasting accuracy without first defining phylogenetic clades, a process that can borrow information from the future or change clade definitions between initial and future timepoints. We identified the best forecasting models as those that minimized this distance between populations. The most accurate sequence-only model for the 12-month forecast horizon estimated fitness with local branching index (LBI) (*Neher et al., 2014*) and mutational load (*Luksza and Lässig, 2014*). As a positive control, we calculated the post hoc empirical fitness of each initial population based on the composition of the corresponding future population. These empirical fitnesses provided the lower bound on the earth mover's distance that represented the number of AA substitutions accumulated between populations.

To understand the effects of reducing forecast horizons and submission lags on long-term forecast accuracy, we produced forecasts 3, 6, 9, and 12 months into the future using HA sequences available at each initial timepoint under each submission lag scenario including no lag, ideal lag (~1-month average), and realistic lag (~3-month average) (*Figure 1*, *Figure 1—figure supplements 2 and 3*). For both natural and simulated populations, we assigned ideal and realistic lags to each sequence from the modeled distributions in *Figure 1B*. This approach allowed us to assign uncorrelated lag values to both population types while avoiding the biases associated with historical submission patterns for natural A/H3N2 HA sequences. For natural A/H3N2 populations, we used the best sequence-only forecasting model, LBI and mutational load, which we previously trained on 12-month forecasts without any submission lag. For simulated A/H3N2-like populations, we used the observed fitness per sample provided by the simulator. For each forecast horizon and submission lag type, we calculated the earth mover's distance between the predicted future populations under the given lag scenario and the observed future populations without any lag in sequence availability. As a control, we also calculated the optimal distance between initial and future populations based on post hoc empirical fitness of the initial population. We anticipated that reducing either the forecast horizon or the submission lag would reduce the distance to the future in AAs, representing increased accuracy of the forecasting models.

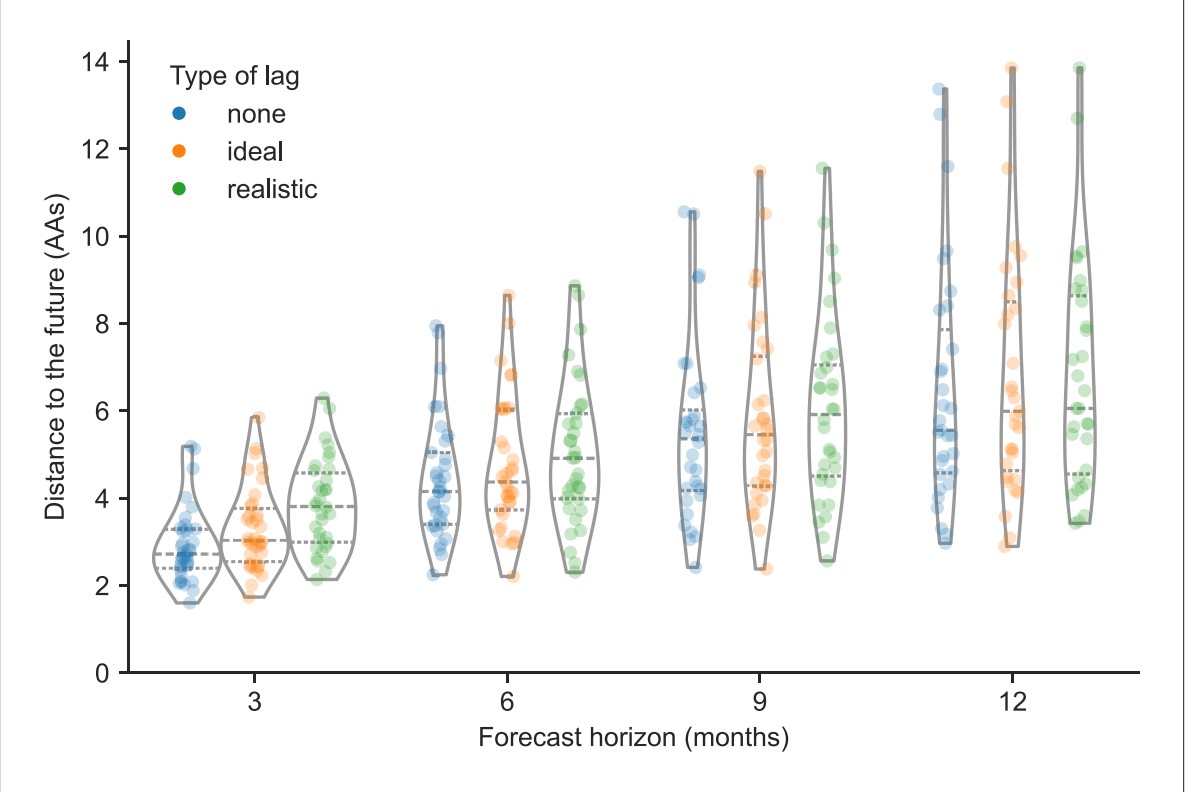

**Figure 2.** Distance to the future per timepoint (AAs) for natural A/H3N2 populations by forecast horizon and submission lag type based on forecasts from the local branching index (LBI) and mutational load model. Each point represents a future timepoint whose population was predicted from the number of months earlier corresponding to the forecast horizon. Points are colored by submission lag type including forecasts made with no lag (blue), an ideal lag (orange), and a realistic lag (green).

The online version of this article includes the following source data, source code, and figure supplement(s) for figure 2:

**Source data 1.** Distance to the future for natural A/H3N2 populations; see h3n2_distances_to_the_future.csv at https://doi.org/10.5281/zenodo.17259448.

**Source code 1.** Jupyter notebook used to produce this figure and the figure supplement: workflow/notebooks/plot-distances-to-the-future-by-delay-type-and-horizon-for-population.py.ipynb.

**Figure supplement 1.** Distance to the future for simulated A/H3N2-like populations by forecast horizon and submission lag type based on forecasts from the "true fitness" model.

**Figure supplement 1—source data 1.** Distance to the future for simulated A/H3N2-like populations; see simulated_distances_to_the_future.csv at https://doi.org/10.5281/zenodo.17259448.

**Figure supplement 2.** Optimal distance to the future for natural A/H3N2 populations by forecast horizon and submission lag type based on post hoc empirical fitness of the initial population.

**Figure supplement 3.** Optimal distance to the future for simulated A/H3N2-like populations by forecast horizon and submission lag type based on post hoc empirical fitness of the initial population.

We found that reducing the forecast horizon from the current standard of 12 months linearly reduced the distance to the future population predicted by the LBI and mutational load model (*Figure 2*). Under all three submission lag scenarios, the distance to the future reduced by approximately 1 AA on average for each 3-month reduction in forecast horizon (*Table 1*). We observed the greatest average reduction in distance to the future (~1.4 AAs) between the 6- and 3-month forecast horizons. Reducing the forecast horizon also noticeably reduced the variance per timepoint in predicted future populations across all lag scenarios (*Figure 2*). For example, the standard deviation of distances to the future reduced from ~2.6 AAs at the 12-month horizon to ~1 AA at the 3-month horizon (*Table 1*). We observed the same patterns for forecasts of simulated A/H3N2-like populations (*Figure 2—figure supplement 1*) and optimal distances to the future for natural and simulated

**Table 1.** Distance to the future in amino acids (mean ± SD AAs) by forecast horizon (in months) and submission lag for A/H3N2 populations.

| | Distance to future (mean ± SD AAs) | | |
|---|---|---|---|
| Horizon | No lag | Ideal lag | Realistic lag |
| 3 | 2.91± 0.86 | 3.32±0.96 | 3.85±1.05 |
| 6 | 4.44±1.39 | 4.74±1.54 | 5.03±1.66 |
| 9 | 5.48± 2.05 | 5.84±2.14 | 6.04±2.15 |
| 12 | 6.45±2.72 | 6.77±2.80 | 6.78±2.61 |

populations (*Figure 2—figure supplements 2 and 3*). Thus, reducing how far we have to predict into the future increased both forecast accuracy and precision.

In contrast, we found that reducing submission lags from a ~3-month average lag in the realistic scenario to a ~1-month average lag in the ideal scenario had a weaker effect on distance to the future. At the 12-month forecast horizon, the ideal and realistic lag scenarios produced similar predictions, with the only noticeable improvement observed under the scenario without any submission lags (*Figure 2*). As the forecast horizon decreased, the effect of submission lags appeared more prominent, with the greatest effect of reduced lags observed at the 3-month forecast horizon. However, the average improvement from the realistic to the ideal submission lag scenario at the 3-month horizon was still only ~0.3 AAs (*Table 1*). Reducing submission lags also had little effect on the variance per timepoint in predicted future populations. Interestingly, we observed a stronger effect of reducing submission lags in simulated A/H3N2-like populations, with the best average improvement between realistic and ideal lags of ~0.7 AAs at the 3-month horizon (*Figure 2—figure supplement 1*). As with natural A/H3N2 populations, the effect of reducing submission lags appeared to increase as the forecast horizon decreased. These results indicate that reducing submission lags may have little effect under the current 12-month forecast approach used for influenza vaccine composition, but reducing submission lags should become increasingly important as we forecast from closer to future influenza populations.

## Reducing submission lags improves estimates of current clade frequencies

Although the distance between predicted and observed future populations in AAs provides an unbiased metric to optimize forecasting models, in practice, we use these models to forecast clade frequencies. We predict each clade's future frequency as the sum of predicted future frequencies for each HA sequence in the clade. We calculate these sequence-specific future frequencies as the initial sequence frequency times the estimated sequence fitness (*Luksza and Lässig, 2014*; *Huddleston et al., 2020*). Given the importance of initial clade frequencies in these forecasts, we tested the effect of submission lags on current clade frequency estimates. For each timepoint and clade with a frequency greater than zero under the scenario without lags, we calculated the clade frequency error as the difference between clade frequency without submission lags and the frequency with either an ideal or realistic lag. Positive error values represented underestimation of current clades, while negative values represented overestimation.

Across all clade frequencies, we found that errors in current clade frequencies for A/H3N2 appeared normally distributed with lower variance in the ideal lag scenario than under realistic lags (*Figure 3A and B*). Of the 822 clades under the scenario without lags, 613 (75%) had a frequency less than 10%, representing small, emerging clades. The remaining 209 (25%) had a frequency of 10% or greater, representing larger clades that could be more likely to succeed. To understand whether lags had different effects on these small and large clades, respectively, we inspected clades from these latter two groups separately. For small clades, errors under ideal lags ranged from –4% to 4% with a standard deviation of 1%, while realistic lags produced errors ranging from –8% to 7% with a standard deviation of 2% (*Figure 3C*). We did not observe a bias toward underestimation or overestimation of initial small clade frequencies under either lag scenario. For large clades, errors under ideal lag ranged from –9% to 14% with a standard deviation of 3% (*Figure 3D*). Errors under realistic lags ranged from –16%

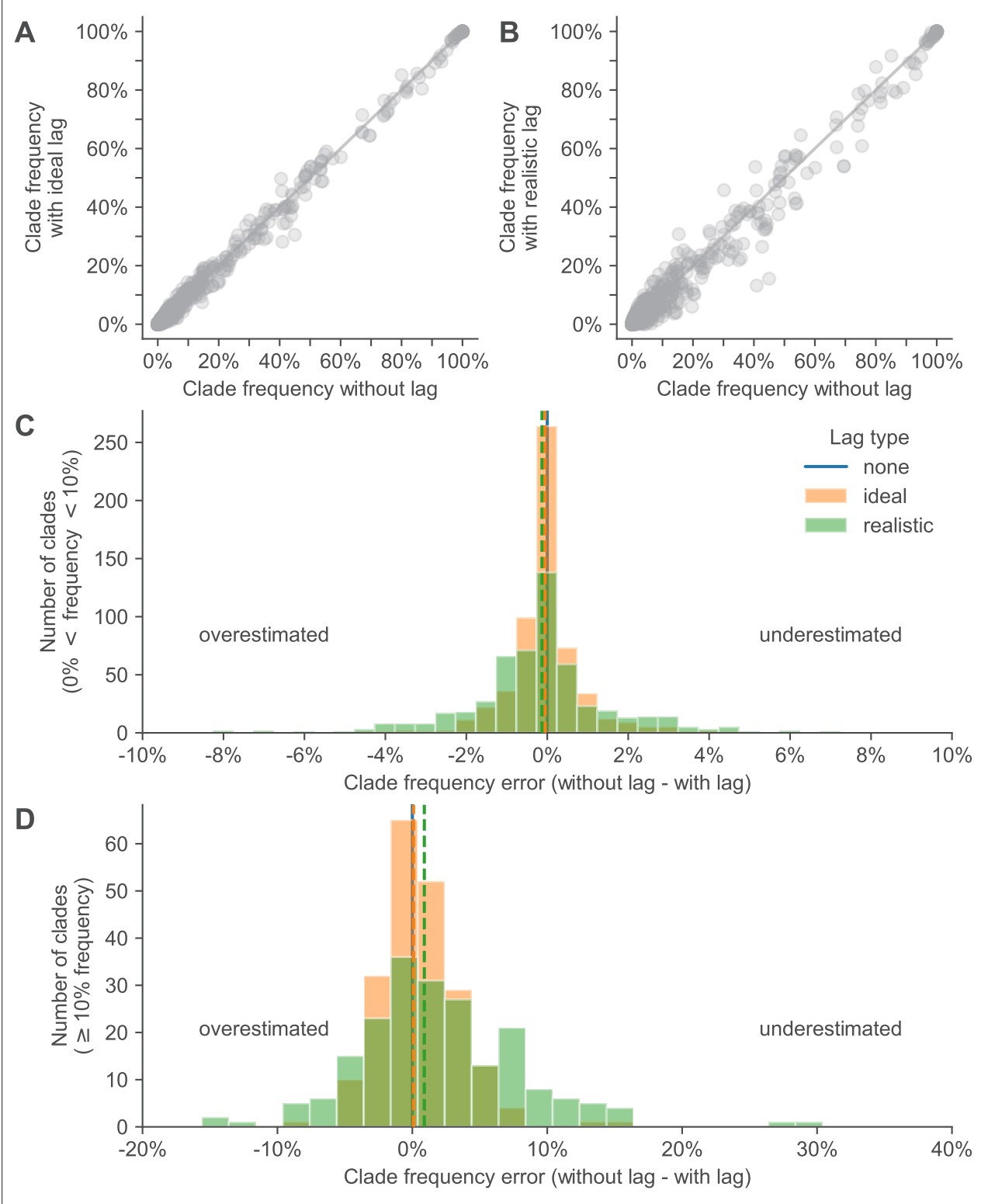

**Figure 3.** Clade frequency errors for natural A/H3N2 clades. Clade frequency errors for natural A/H3N2 clades at the same timepoint calculated as the difference between clade frequencies without submission lag and corresponding frequencies with either (**A**) ideal or (**B**) realistic submission lags. Distributions of frequency errors appear normally distributed in both lag scenarios for both (**C**) small clades (>0% and <10% frequency) and (**D**) large clades (≥10%). Dashed lines indicate the median error from the distribution of the lag type with the same color.

The online version of this article includes the following source data, source code, and figure supplement(s) for figure 3:

*Figure 3 continued on next page*

*Figure 3 continued*

**Source data 1.** Current and future clade frequencies for natural A/H3N2 populations by forecast horizon and submission lag type; see h3n2_clade_ frequencies.csv at https://doi.org/10.5281/zenodo.17259448.

**Source code 1.** Jupyter notebook used to produce this figure and the figure supplement: workflow/notebooks/plot-current-clade-frequency-errors-by-delay-type-for-populations.py.ipynb.

**Figure supplement 1.** Clade frequency errors between simulated A/H3N2-like HA populations with ideal or realistic submission lags and populations without any submission lag.

**Figure supplement 1—source data 1.** Current and future clade frequencies for simulated A/H3N2-like populations by forecast horizon and submission lag type; see simulated_clade_frequencies.csv athttps://doi.org/10.5281/zenodo.17259448.

to 29% with a standard deviation of 6%. We observed a slight bias toward underestimation of large clades under the realistic lag scenario, with a median error of 1%. These results show that reducing submission lags for natural A/H3N2 populations from a 3-month average to a 1-month average could reduce the bias toward underestimated large clade frequencies and reduce the standard deviation of all current clade frequency errors by 50%.

Lagged submissions similarly affected clade frequencies for simulated A/H3N2-like populations (*Figure 3—figure supplement 1*). Small clade errors under ideal lags ranged from –4% to 6% (standard deviation of 1%) and under realistic lags ranged from –9% to 8% (standard deviation of 2%) (*Figure 3—figure supplement 1C*). For large clades, errors under ideal lags ranged from –8% to 18% (standard deviation of 3%) and under realistic lags from –14% to 40% (standard deviation of 7%) (*Figure 3—figure supplement 1D*). As with natural A/H3N2 populations, we observed a slight bias in simulated populations under realistic lags toward underestimation of large clade frequencies with a median error of 2%. We also observed a similar reduction in standard deviation of current frequency errors for these simulated A/H3N2-like populations when switching from realistic to ideal submission lags.

## Reducing forecast horizons increases the accuracy and precision of clade frequency forecasts

Next, we estimated the effects of different forecast horizons and submission lags on the accuracy of clade frequency forecasts. As with the current clade frequency analysis, we analyzed small clades (<10% initial frequency) and large clades (≥10% initial frequency) separately. For each combination of initial timepoint, future timepoint, and lag scenario (*Figure 1*), we calculated initial and predicted future frequencies for all clades present under the given lag and then calculated the corresponding observed future frequencies without lag for clades that descended from the clades present at the initial timepoint. We calculated the error in forecast frequencies as the difference between predicted future frequencies under the given lag scenario and observed future frequencies without any lag. We used absolute forecast errors to evaluate forecast accuracy and overall forecast errors to evaluate forecast bias.

Absolute forecast errors trended strongly toward values less than 30% with long tails reaching 80% for both small and large clades (*Figure 4*). Each 3-month reduction of the forecast horizon linearly reduced the variance in forecast errors, but mean and median absolute errors only improved after reducing the forecast horizon below 9 months (*Figure 4* and *Table 2*). For small clades, reducing the forecast horizon most noticeably reduced the range of errors, while reducing submission lags had little effect (*Figure 4A*). For large clades, almost all decreases in forecast horizon and submission lag (except lags at the 12-month horizon) reduced the standard deviation of absolute forecast errors (*Figure 4B*). Overall, reducing the forecast horizon had a greater effect on the mean, median, and standard deviation of absolute forecast errors than reducing submission lags. For example, the standard deviation of absolute errors at the 12-month horizon under realistic submission lags was 23%, while the standard deviation for the 6-month horizon under realistic lags was 14% (*Table 2*). In contrast, the standard deviation at the 12-month horizon under ideal submission lags did not change from the realistic lags at 23%, and the average absolute error increased by 1% from 20%. For all other forecast horizons, reducing the submission lags from realistic to ideal only reduced the mean and standard deviation of absolute errors by 1–2%. We observed the same general patterns in simulated populations (*Figure 4—figure supplement 1*).

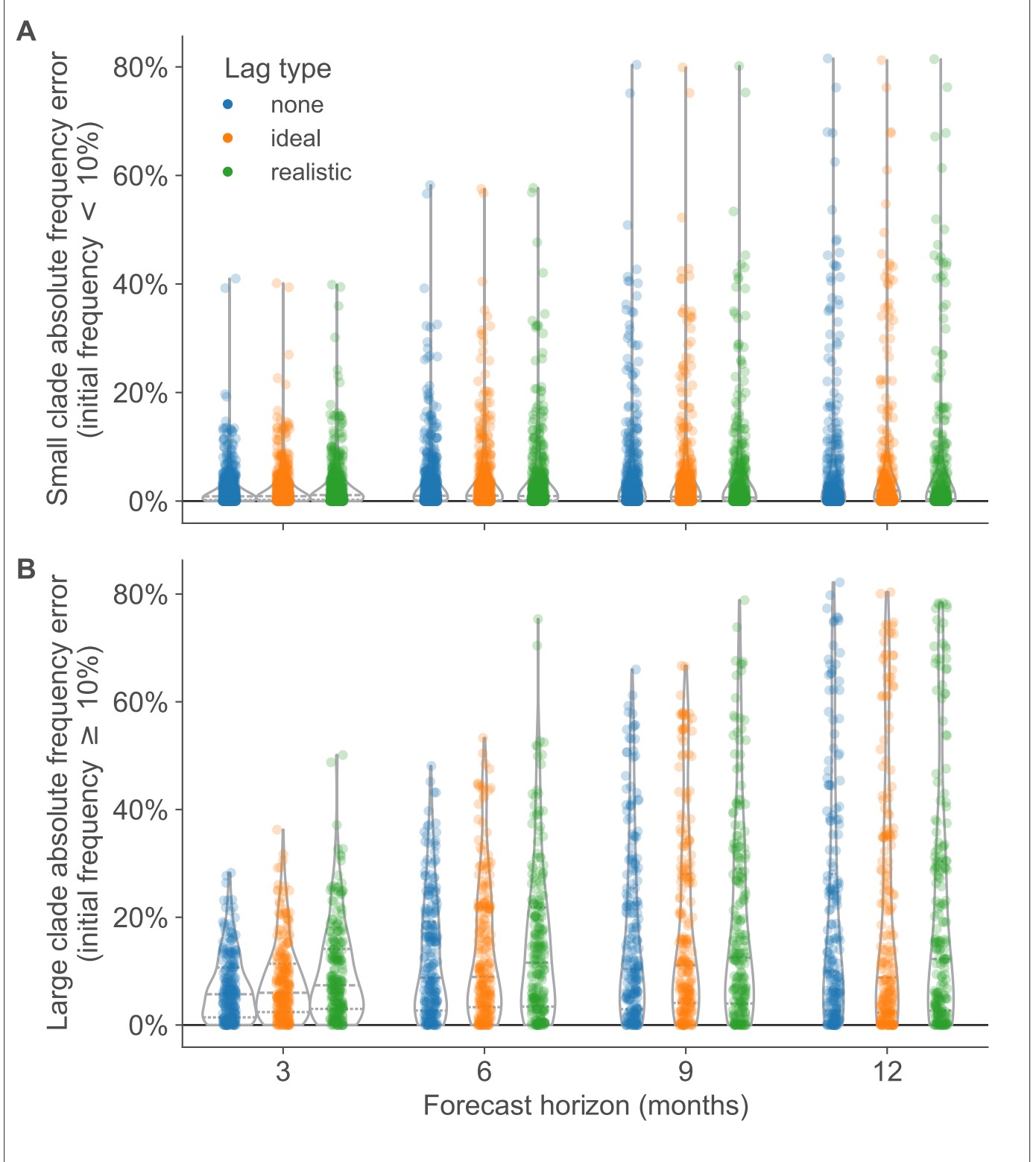

**Figure 4.** Absolute forecast clade frequency errors for natural A/H3N2 populations by forecast horizon in months and submission lag type (none, ideal, or observed) for (**A**) small clades (<10% initial frequency) and (**B**) large clades (≥10% initial frequency).

The online version of this article includes the following source code and figure supplement(s) for figure 4:

**Source code 1.** Jupyter notebook used to produce this figure and the figure supplements: workflow/notebooks/plot-forecast-clade-frequency-errors-by-delay-type-and-horizon-for-population.py.ipynb.

*Figure 4 continued on next page*

*Figure 4 continued*

**Figure supplement 1.** Absolute forecast clade frequency errors for simulated A/H3N2-like HA populations by forecast horizon in months and submission lag type (none, ideal, or realistic) for (**A**) small clades (<10% initial frequency) and (**B**) large clades (≥10% initial frequency).

**Figure supplement 2.** Forecast clade frequency errors for natural A/H3N2 HA populations by forecast horizon in months and submission lag type (none, ideal, or realistic) for (**A**) small clades (10% initial frequency) and (**B**) large clades (≥10% initial frequency).

**Figure supplement 3.** Forecast clade frequency errors for simulated A/H3N2-like HA populations by forecast horizon in months and submission lag type (none, ideal, or realistic) for (**A**) small clades (<10% initial frequency) and (**B**) large clades (≥10% initial frequency).

The majority of forecast frequency errors appeared to be normally distributed, indicating little bias toward over- or underestimating future clade frequencies (*Figure 4—figure supplements 2 and 3*). This pattern matched our expectation that at any given initial timepoint the overestimation of one clade's future frequency must cause an underestimation of another current clade's future frequency. However, we observed a long tail of small clades with underestimated future frequencies at all forecast horizons, indicating that correctly predicting the growth of small clades remains more difficult than predicting their decline (*Figure 4—figure supplement 2A*). The strongest effect of reducing submission lags was the reduction in maximum error, corresponding to reduction in underestimation of large clades. The switch from realistic to ideal lags at 12-, 9-, 6-, and 3-month horizons reduced the maximum forecast error by 4%, 21%, 22%, and 14%, respectively (*Table 2*). These results show that reducing submission lags can substantially lower the upper bound for forecasting errors.

## Reduced vaccine development time provides the best improvement in forecast accuracy of available realistic interventions

Although we have investigated the effects of a range of forecast horizons and submission lags, not all of these scenarios are currently realistic. The most we can hope to reduce the forecast horizon with current mRNA vaccine technology is from 12 months to 6 months, and the most we could reduce submission lags would be from an average of 3 months to 1 month (*Grant et al., 2023*). In practice, we wanted to know how much a reduction in forecast horizon or submission lag could improve the accuracy of forecasts to each future timepoint. To determine the effects of realistic interventions on forecast accuracy, we inspected the reduction in total absolute forecast error per future timepoint associated with improved vaccine development (reducing forecast horizon from 12 months to 6 months), improved genomic surveillance (reducing lags from a 3-month average to 1 month), and the combination of both improvements. We selected all forecasts with a 12-month

**Table 2.** Errors in clade frequencies between observed and predicted values by forecast horizon (in months) and submission lag for A/H3N2 clades with an initial frequency ≥10% under the given lag scenario.

| Horizon | Lag type | Clade frequency error (%) | | | | | Absolute frequency error (%) | | |
| --- | --- | --- | --- | --- | --- | --- | --- | --- | --- |
| | | Mean | Median | SD | Min | Max | Mean | Median | SD |
| 3 | None | 1 | 0 | 9 | –28 | 28 | 7 | 6 | 6 |
| 3 | Ideal | 1 | 0 | 11 | –32 | 36 | 8 | 6 | 7 |
| 3 | Realistic | 1 | 0 | 13 | –31 | 50 | 10 | 7 | 9 |
| 6 | None | 1 | 0 | 17 | –48 | 45 | 12 | 9 | 11 |
| 6 | Ideal | 1 | 0 | 19 | –50 | 53 | 13 | 9 | 13 |
| 6 | Realistic | 1 | 0 | 20 | –52 | 75 | 15 | 12 | 14 |
| 9 | None | 0 | -1 | 23 | –66 | 59 | 16 | 10 | 17 |
| 9 | Ideal | 1 | -1 | 25 | –67 | 58 | 18 | 11 | 18 |
| 9 | Realistic | 1 | -1 | 26 | –67 | 79 | 19 | 12 | 19 |
| 12 | None | 0 | 0 | 30 | –82 | 76 | 20 | 10 | 22 |
| 12 | Ideal | 1 | 0 | 31 | –80 | 74 | 21 | 9 | 23 |
| 12 | Realistic | 0 | 0 | 31 | –78 | 78 | 20 | 12 | 23 |

horizon and a realistic lag to represent current forecast conditions or "the status quo". For the same future timepoints present in the status quo conditions, we selected the corresponding forecasts for a 6-month horizon and a realistic lag, a 12-month horizon and an ideal lag, and 6-month horizon and an ideal lag. Since forecasts between different initial and future timepoints could be represented by different clades, we could not compare forecasts for specific clades between interventions. Instead, we calculated the total absolute clade frequency error per future timepoint under each intervention and calculated the improvement in forecast accuracy as the difference in total error between the status quo and each intervention. In addition to this clade-based analysis, we also estimated effects of interventions on the difference in distance to the future between different scenarios for both estimated and empirical fitnesses. For all analyses, positive values represented improved forecast accuracy under a given intervention scenario and negative values represented a reduction in accuracy.

Both interventions with improved vaccine development increased forecast accuracy for the majority of future timepoints (*Figure 5*, *Table 3*, and *Figure 5—figure supplement 1*). Improving vaccine development alone increased total forecast accuracy by 53% on average, while the addition of improved genomic surveillance under that 6-month forecast horizon increased total forecast accuracy by 54% on average. In contrast, the intervention that only improved genomic surveillance decreased forecast accuracy by an average of 11%. Based on the distributions of total absolute forecast error per future timepoint, we would expect improved genomic surveillance to improve forecast accuracy at a forecast horizon of 3 months (*Figure 5—figure supplement 1*). We observed similar effects of interventions in simulated A/H3N2-like populations, except that the average effect of reducing submission lags alone was positive for these populations (*Figure 5—figure supplements 2 and 3*). When we calculated the effects of interventions on distances to the future instead of total absolute clade frequency errors, we observed the same patterns for natural and simulated populations (*Figure 5—figure supplements 4 and 5*). Based on these results, the single most valuable intervention we could make to improve forecast accuracy would be to reduce the forecast horizon to 6 months or less through more rapid vaccine development. However, as we reduce the forecast horizon, reducing submission lags should have a greater effect on improving forecast accuracy.

We hypothesized that the decrease in average accuracy of natural A/H3N2 forecasts under the improved genomic surveillance intervention could reflect the bias of the LBI and mutational load fitness metrics. For example, we previously showed how LBI fitness estimates can overestimate the future growth of large clades (*Huddleston et al., 2020*). Adding more sequences at initial timepoints where LBI already overestimates clade success could increase the LBI of those clades and exacerbate the overestimation. To test this hypothesis, we calculated the effects of the same interventions on the optimal distances to the future for both natural and simulated populations. Since optimal distances reflected the empirical fitnesses of the initial populations, the effects of interventions should be independent of biases from fitness metrics. We expected all interventions to maintain or improve the optimal distance to the future without any cases where an intervention decreased accuracy.

As expected, all interventions improved on the optimal distance to the future for both populations (*Figure 6* and *Figure 6—figure supplement 1*). For natural A/H3N2 populations, the average improvement of the vaccine intervention was 1.1 AAs and the improvement of the surveillance intervention was 0.27 AAs or approximately 25% of the vaccine intervention. The average improvement of both interventions was only slightly less than additive at 1.28 AAs. To verify the robustness of these results, we replicated our entire analysis of A/H3N2 populations using a subsampling scheme that tripled the number of viruses selected per month from 90 to 270 (*Figure 1—figure supplement 4C*). We found the same pattern with this replication analysis, with average improvements of 0.93 AAs for the vaccine intervention, 0.21 AAs for the surveillance intervention, and 1.14 AAs for both interventions (*Figure 6—figure supplement 2*). These effects of realistic interventions appeared consistent across the range of genetic diversity at future timepoints (*Figure 6—figure supplement 3*) and for future seasons occurring in both Northern and Southern Hemispheres (*Figure 6—figure supplement 4*). We noted a slightly greater median improvement in forecast accuracy associated with both improved vaccine interventions for the Southern Hemisphere seasons (1.03 and 1.42 AAs) compared to the Northern Hemisphere seasons (0.74 and 0.93 AAs). These results confirmed the relatively stronger effect of reducing forecast horizons compared to submission lags. They also confirmed that reducing submission lags can improve forecasts under optimal forecasting conditions. For this

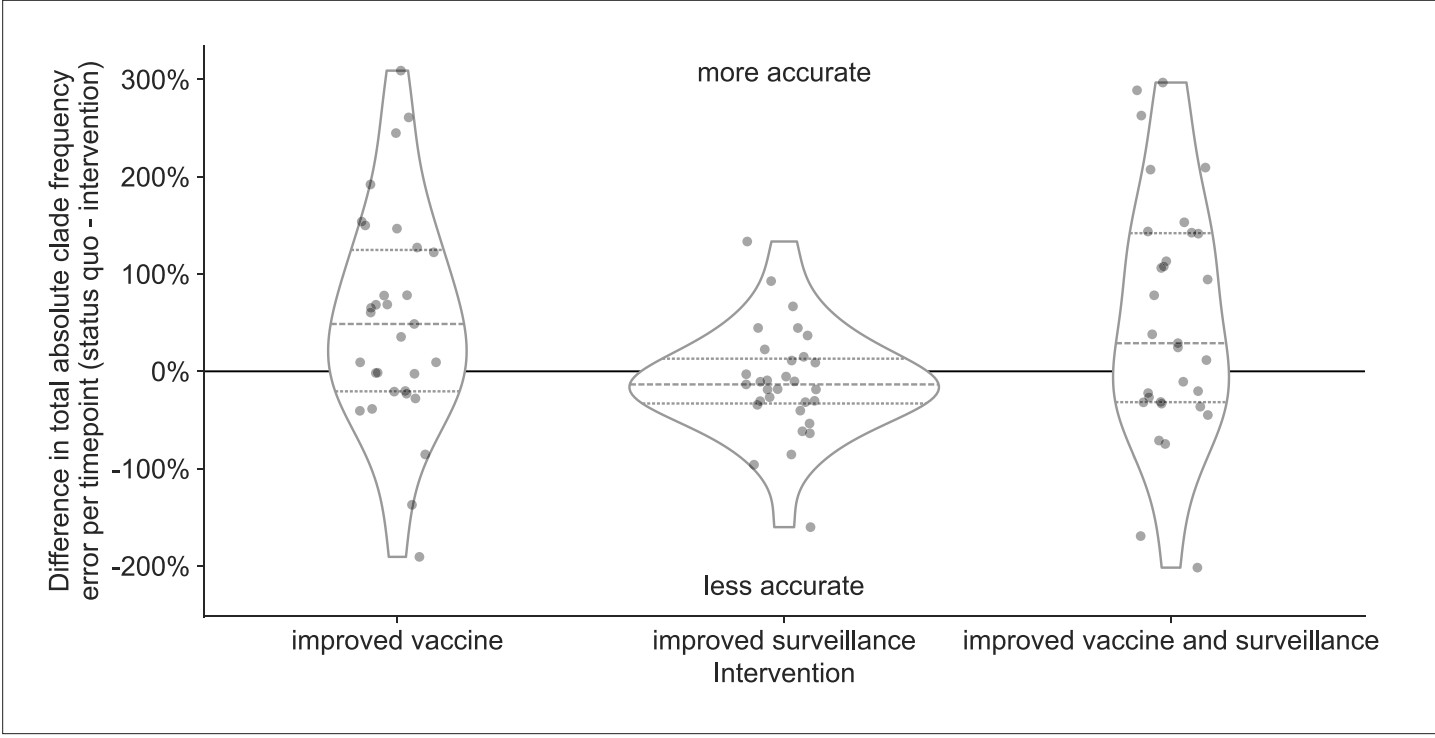

**Figure 5.** Improvement of clade frequency errors for A/H3N2 populations between the status quo (12-month forecast horizon and realistic submission lags) and realistic interventions of improved vaccine development (reducing 12-month to 6-month forecast horizon), improved surveillance (reducing submission lags from 3 months on average to 1 month), or a combination of both interventions. We measured improvements from the status quo as the difference in total absolute clade frequency error per future timepoint. Positive values indicate increased forecast accuracy, while negative values indicate decreased accuracy. Each point represents the improvement of forecasts for a specific future timepoint under the given intervention. Horizontal dashed lines indicate median improvements. Horizontal dotted lines indicate upper and lower quartiles of improvements.

The online version of this article includes the following source data, source code, and figure supplement(s) for figure 5:

**Source data 1.** Differences in total absolute clade frequency error per future timepoint and clade between the status quo and realistic interventions for A/H3N2 populations; see h3n2_effects_of_realistic_interventions.csv at https://doi.org/10.5281/zenodo.17259448.

**Source code 1.** Jupyter notebook used to produce effects of interventions on total absolute clade frequency errors workflow/notebooks/plot-forecast-clade-frequency-errors-by-delay-type-and-horizon-for-population.py.ipynb.

**Source code 2.** Jupyter notebook used to produce effects of interventions on distances to the future: workflow/notebooks/plot-distances-to-the-future-by-delay-type-and-horizon-for-population.py.ipynb.

**Figure supplement 1.** Distribution of total absolute clade frequency errors summed across clades per future timepoint for A/H3N2 populations.

**Figure supplement 2.** Improvement of clade frequency errors for simulated A/H3N2-like populations between the status quo and realistic interventions.

**Figure supplement 2—source data 1.** Differences in total absolute clade frequency error per future timepoint and clade between the status quo and realistic interventions for simulated A/H3N2-like populations; see simulated_effects_of_realistic_interventions.csv at https://doi.org/10.5281/zenodo.17259448.

**Figure supplement 3.** Distribution of total absolute clade frequency errors summed across clades per future timepoint for simulated A/H3N2-like populations.

**Figure supplement 4.** Improvement of distances to the future (AAs) for A/H3N2 populations between the status quo (12-month forecast horizon and realistic submission lags) and realistic interventions.

**Figure supplement 4—source data 1.** Improvement of distances to the future per future timepoint for A/H3N2 populations; see h3n2_effects_of_realistic_interventions_on_distances_to_the_future.csv at https://doi.org/10.5281/zenodo.17259448.

**Figure supplement 5.** Improvement of distances to the future (AAs) for simulated A/H3N2-like populations between the status quo (12-month forecast horizon and realistic submission lags) and realistic interventions.

**Figure supplement 5—source data 1.** Improvement of distances to the future per future timepoint for simulated A/H3N2-like populations; see simulated_effects_of_realistic_interventions_on_distances_to_the_future.csv at https://doi.org/10.5281/zenodo.17259448.

**Table 3.** Improvement in A/H3N2 clade frequency forecast accuracy under realistic interventions of improved vaccine development (reducing 12-month to 6-month forecast horizon), improved surveillance (reducing submission lags from 3 months on average to 1 month), or a combination of both interventions.

We measured improvements from the status quo (12-month forecast horizon and 3-month average submission lag) as the difference in total absolute clade frequency error per future timepoint and the number and proportion of future timepoints for which forecasts improved under the intervention.

| | Forecast accuracy improvement (%) | | | Timepoints improved | |
|---|---|---|---|---|---|
| Intervention | Mean | Median | SD | Total | Proportion |
| Improved vaccine | 53 | 49 | 112 | 19 | 0.61 |
| Improved surveillance | −11 | −13 | 56 | 10 | 0.32 |
| Improved vaccine and surveillance | 54 | 29 | 124 | 18 | 0.58 |

reason, we expect that simultaneous improvements to forecasting models and genomic surveillance will have a mutually beneficial effect on forecast accuracy.

## Discussion

In this work, we showed that realistic public health policy changes that decrease the time to develop new vaccines for seasonal influenza A/H3N2 and decrease submission lags of HA sequences to public databases could improve our estimates of future and current populations, respectively. We confirmed that forecasts became more accurate and more precise with each 3-month reduction in forecast horizon from the status quo of 12 months. Although decreasing submission lags only marginally improved long-term forecast accuracy, shorter lags increased the accuracy of current clade frequency estimates, reduced the bias toward underestimating current and future frequencies of larger clades, and improved forecasts 3 months into the future. Under a realistic scenario where a shorter vaccine development timeline allowed us to forecast from 6 months before the next season, we found a 53% average improvement in forecasts of total absolute clade frequency and a 25% reduction in average absolute forecast frequency errors for large clades from 20% to 15%. We confirmed these effects with a previously validated forecasting model using both simulated and natural populations and two different metrics of forecast accuracy including earth mover's distances between populations and clade frequencies. Since all models to date rely on currently available HA sequences to determine the clades to be forecasted, we expect that decreasing forecast horizons and submission lags will have similar relative effect sizes across all forecasting models including those that integrate phenotypic and genetic data.

Even without these recommended improvements to vaccine development and sequence submissions, these results inform important next steps to improve forecasting models. Current and future frequency estimates should be presented with corresponding uncertainty intervals. From this work, we know that our current frequency estimates for large clades (≥10% frequency) under realistic submission lags have a wide range of errors (−16% to 29%). Similarly, the range of 12-month forecast frequency errors under realistic lags includes overestimates by up to 78% and underestimates up to 78%. Long-term forecasts with incomplete current data are highly uncertain by their nature. To support informed decisions about vaccine updates, we must communicate that uncertainty of the present and future to decision-makers. One simple immediate strategy to provide these uncertainty estimates is to estimate current and future clade frequencies from count data with multinomial probability distributions.

Another immediate improvement would be to develop models that can use all available data in a way that properly accounts for geographic and temporal biases. For example, virus samples from North America and Europe are overrepresented in the GISAID EpiFlu database, while samples from Africa and Asia are underrepresented (*Figure 1—figure supplement 4*). As new H3N2 epidemics often originate from East and Southeast Asia and burn out in North America and Europe (*Bedford et al., 2015*), models that do not account for this geographic bias are more likely to incorrectly predict the success of lower fitness variants circulating in overrepresented regions and miss higher fitness variants emerging from underrepresented regions. Additionally, the number of H3N2 HA sequences per year in the GISAID EpiFlu database has increased consistently since 2010, creating a temporal bias

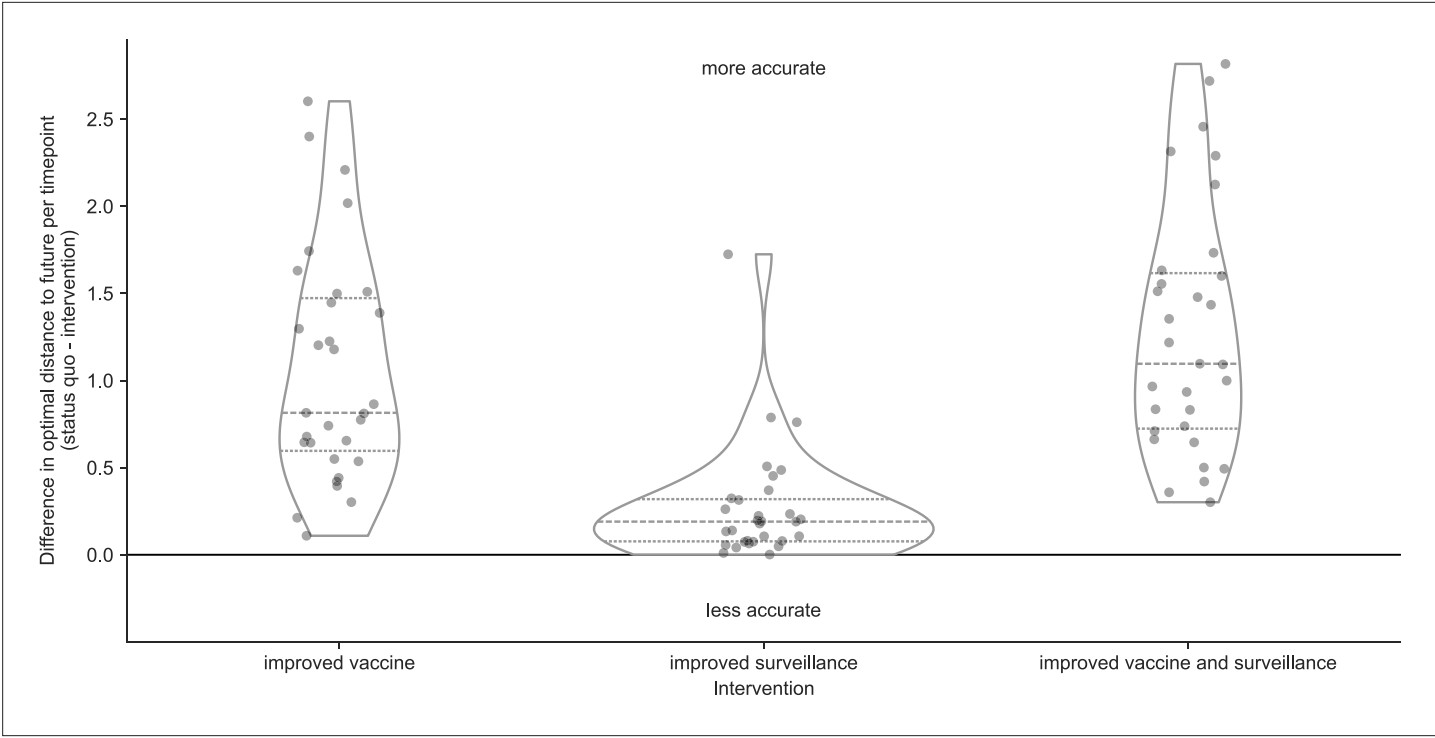

**Figure 6.** Improvement of optimal distances to the future (AAs) for A/H3N2 populations between the status quo (12-month forecast horizon and realistic submission lags) and realistic interventions of improved vaccine development (reducing 12-month to 6-month forecast horizon), improved surveillance (reducing submission lags from 3 months on average to 1 month), or a combination of both interventions. We measured improvements from the status quo as the difference in optimal distances to the future per future timepoint. Positive values indicate increased forecast accuracy, while negative values indicate decreased accuracy. Each point represents the improvement of forecasts for a specific future timepoint under the given intervention. Horizontal dashed lines indicate median improvements. Horizontal dotted lines indicate upper and lower quartiles of improvements.

The online version of this article includes the following source data, source code, and figure supplement(s) for figure 6:

**Source data 1.** Differences in optimal distances to the future per future timepoint between the status quo and realistic interventions for A/H3N2 populations; see h3n2_optimal_effects_of_realistic_interventions_on_distances_to_the_future.csv at https://doi.org/10.5281/zenodo.17259448.

**Source code 1.** Python notebook used to produce optimal effects of interventions on distances to the future: workflow/notebooks/plot-distances-to-the-future-by-delay-type-and-horizon-for-population.py.ipynb.

**Source code 2.** Python notebook used to plot optimal effects by future clade entropy: workflow/notebooks/plot-optimal-effects-of-interventions-by-clade-entropy.py.

**Source code 3.** Python notebook used to plot optimal effects by hemisphere: workflow/notebooks/plot-optimal-effects-of-interventions-by-hemisphere.py.

**Figure supplement 1.** Improvement of optimal distances to the future (AAs) for simulated A/H3N2-like populations between the status quo (12-month forecast horizon and realistic submission lags) and realistic interventions.

**Figure supplement 1—source data 1.** Improvement of optimal distances to the future per future timepoint for simulated A/H3N2-like populations; see simulated_optimal_effects_of_realistic_interventions_on_distances_to_the_future.csv at https://doi.org/10.5281/zenodo.17259448.

**Figure supplement 2.** Improvement of optimal distances to the future (AAs) for A/H3N2 populations between the status quo (12-month forecast horizon and realistic submission lags) and realistic interventions using forecasts based on sampling 270 viruses per month instead of the 90 viruses-per-month sampling used in the main results.

**Figure supplement 2—source data 1.** Improvement of optimal distances to the future per future timepoint for A/H3N2 populations with higher density sampling; see h3n2_high_density_optimal_effects_of_realistic_interventions_on_distances_to_the_future.csv at https://doi.org/10.5281/zenodo.17259448.

**Figure supplement 3.** Improvement of optimal distances to the future (AAs) for A/H3N2 populations compared to the Shannon entropy of clade frequencies (estimated without submission lags) at the future timepoint being forecast to.

**Figure supplement 3—source data 1.** Improvement of optimal distances to the future for A/H3N2 populations compared to the Shannon entropy of clade frequencies at the future timepoint; see h3n2_optimal_effects_of_realistic_interventions_on_distances_to_the_future_by_future_clade_entropy.csv at https://doi.org/10.5281/zenodo.17259448.

*Figure 6 continued on next page*

*Figure 6 continued*

**Figure supplement 4.** Improvement of optimal distances to the future (AAs) for A/H3N2 populations by intervention and the hemisphere with an active season during the future timepoint being predicted.

where any given season a model forecasts to will have more sequences available than the season from which forecasts occur. The model we used in this study does not explicitly account for geographic variability of viral fitness and relies on time-scaled phylogenetic trees which can be computationally costly to infer for large sample sizes. As a result, we needed to evenly sample the diversity of currently circulating viruses to produce unbiased trees in a reasonable amount of time. Models that could estimate viral fitness per geographic region without inferring trees could use more available sequence data and reduce the uncertainty in current and future clade frequencies.

Finally, we could improve existing models by changing the start and end times of our long-term forecasts. We could change our forecasting target from the middle of the next season to the beginning of the season, reducing the forecast horizon from 12 to 9 months. We could also start forecasting from 1 month prior to the current date to minimize the effect of submission lags on our estimates of the current global influenza population.

Despite the small effect that reducing sequence submission lags had on long-term forecasting accuracy, we still see a need to continue funding global genomic surveillance at higher levels than the pre-pandemic period. Compared to estimates of current viral diversity, forecasts of future influenza populations only represent one component of the overall decision-making process for vaccine development. For example, virologists must choose potential vaccine candidates from the diversity of circulating clades months in advance of vaccine composition meetings to have time to grow virus in cells and eggs and measure antigenic drift with serological assays (*Morris et al., 2018*; *Loes et al., 2024*). Earlier detection of viral sequences with important antigenic substitutions could determine whether corresponding vaccine candidates are available at the time of the vaccine selection meeting or not. Newer methods to estimate influenza fitness use experimental measurements of viral escape from human sera (*Lee et al., 2019*; *Welsh et al., 2024*; *Meijers et al., 2025*; *Kikawa et al., 2025*), measurements of viral stability and cell entry (*Yu et al., 2025*), or sequences from neuraminidase, the other primary surface protein associated with antigenic drift (*Meijers et al., 2025*). These methodological improvements all depend fundamentally on timely genomic surveillance efforts and the GISAID EpiFlu database to identify relevant influenza variants to include in their experiments. Finally, our results here reflect uncorrelated submission lags for each sequence, but actual lags can strongly correlate between sequences from the same originating and submitting labs. These correlated lags could further decrease the accuracy of frequency estimates beyond our more conservative estimates. More rapid sequence submission will improve our understanding of the present and give decision-makers more choices for new vaccines. Such reductions in submission lags depend on substantial, sustained funding and capacity building globally.

## Materials and methods
### Selection of natural influenza A/H3N2 HA sequences

We downloaded all A/H3N2 HA sequences and metadata from GISAID's EpiFlu database (*Shu and McCauley, 2017*) as of November 2023. We evenly sampled sequences geographically and temporally as previously described (*Huddleston et al., 2020*). Briefly, we selected 90 sequences per month, evenly sampling from major continental regions (Africa, Europe, North America, China, South Asia, Japan and Korea, Oceania, South America, Southeast Asia, and West Asia) and excluding sequences labeled as egg-passaged or missing complete date annotations. For our forecasting analyses, we selected sequences collected between April 1, 2005 and October 1, 2019. This sampling approach accounts for known regional biases in sequence availability through time (*McCarron et al., 2022*) and makes inference of divergence and time trees computationally tractable. This approach also exactly matches our previous study where we first trained the forecast models used in this study (*Huddleston et al., 2020*), allowing us to reuse those previously trained models. With this subsampling approach, we selected between 7% (Europe) and 91% (Southeast Asia) of all available sequences per region across the entire study period with an average of 50% and median of 52% across all 10 regions (*Figure 1—figure supplement 4*). To verify the reproducibility and robustness of our results, we reran

the full forecasting analysis with a high-density subsampling scheme that selected 270 sequences per month with the same even sampling across regions and time as the original scheme. With this approach, we selected between 17% (Europe) and 97% (Southeast Asia) of all available sequences per region with an average of 72% sampled and a median of 83% (*Figure 1—figure supplement 4C*).

## Simulation of influenza A/H3N2-like HA sequences

We simulated A/H3N2-like populations as previously described (*Huddleston et al., 2020*). Briefly, we simulated A/H3N2 HA sequences with SANTA-SIM (*Jariani et al., 2019*) for 10,000 generations or 50 years at 200 generations per year. We discarded the first 10 years of simulated data as a burn-in period and used the next 30 years of the remaining data for our analyses. We sampled 90 viruses per month to match the sampling density of natural populations.

## Estimating and assigning submission lags

We estimated the lag between sample collection and submission of A/H3N2 HA sequences to the GISAID EpiFlu database (*Shu and McCauley, 2017*) by calculating the difference in GISAID-annotated submission date and collection date in days for samples collected between January 1, 2019, and January 1, 2020, and with a submission date prior to October 1, 2020. We selected this period of time as representative of modern genomic surveillance efforts prior to changes in circulation patterns of influenza caused by the SARS-CoV-2 pandemic. Of the 104,392 HA sequences in GISAID EpiFlu, 11,222 (11%) were collected during this period with a mean submission lag of 98 days (~3 months) and a median lag of 74 days. Only 11% of sequences (N=1210) were submitted within 4 weeks of collection, and only 36% (N=4057) were submitted within 8 weeks (*Figure 1A*, purple).

We modeled the shape of the observed lag distribution as a gamma distribution using a maximum likelihood fit from SciPy 1.10.1 (*Virtanen et al., 2020*). With this approach, we estimated a shape parameter of 1.76, a scale parameter of 53.18, and a location parameter of 3.98. The product of these shape and scale values corresponded to a mean lag of 93.76 days (*Figure 1A*, green). To assign realistic submission lags to each sample in our analysis, we randomly sampled from this gamma distribution and calculated a "realistic submission date" by adding the sampled lag in days to the observed collection date. This approach allowed us to assign realistic lags to natural and simulated populations without the biases and autocorrelations associated with historical submission patterns across different submitting labs.

Based on the observed rapid submission of SARS-CoV-2 genomes during the first years of the pandemic, we expected that an achievable "ideal" submission lag for seasonal influenza sequences would have a 1-month average lag instead of the observed ~3-month lag from the pre-pandemic period. We modeled this ideal submission lag distribution by dividing the gamma shape parameter by 3 to get a value of 0.59 and a corresponding mean lag of 31.25 days (*Figure 1A*, orange). This approach effectively shifted the realistic gamma toward zero, while maintaining the relatively longer upper tail of the distribution. To assign ideal submission lags to each sample in our analysis, we randomly sampled from this modified gamma distribution and added the sampled lag in days to the observed collection date. Additionally, we required that each sample's "ideal" lag be less than or equal to its "realistic" lag.

To estimate the effect of increased global sequencing capacity associated with the response to the SARS-CoV-2 pandemic, we summarized the lag distribution for sequences submitted to GISAID EpiFlu between January 1, 2022, and January 1, 2023. During this period, global influenza circulation had rebounded to its prepandemic level and 26,394 HA sequences were collected. The mean and median submission lags during this period were 76 and 62 days, respectively, representing a trend toward reduced lags compared to the prepandemic era (*Figure 1—figure supplement 1*).

## Phylogenetic inference

We inferred time-scaled phylogenetic trees for HA sequences as previously described (*Huddleston et al., 2020*). Briefly, we aligned sequences with MAFFT v7.520 (*Katoh et al., 2002*; *Katoh and Standley, 2013*) using the `augur align` command in Augur v22.3.0 (*Huddleston et al., 2021*). We inferred phylogenies with IQ-TREE v2.2.3 (*Nguyen et al., 2015*) using the `augur tree` command with IQ-TREE parameters of `-ninit 2 -n 2 -me 0.05` and a general time reversible (GTR)

model. We inferred time-resolved phylogenies with TreeTime v0.10.1 (*Sagulenko et al., 2018*) with the `augur refine` command.

## Forecasting with different forecast horizons

We tested the effect of forecasting future influenza populations at forecast horizons of 3, 6, 9, and 12 months (*Figure 1B*). Previously, we produced forecasts every 6 months starting from October 1 and April 1 and predicting 12 months into the future (*Huddleston et al., 2020*). To support forecasts in 3-month intervals, we produced annotated time trees for 6 years of HA sequences every 3 months with data available up to the first day of January, April, July, and October. We produced these trees for each timepoint with three different lag scenarios: no lag, ideal lag, and realistic lag. For each scenario, we selected sequences for analysis at a given timepoint based on their collection date, ideal submission date, or realistic submission date, respectively. This experimental design produced forecasts for three lag types at each of the four forecast horizons (e.g., *Figure 1B*, blue, green, and orange initial timepoints for the 3-month forecast horizon).

Since reliable submission dates were not available prior to April 2005, our analysis of natural A/H3N2 sequences spanned from April 1, 2005, to October 1, 2019. To simplify the data required for these analyses, we produced forecasts of natural A/H3N2 populations with our best sequence-only model from our prior work (*Huddleston et al., 2020*), a composite model based on LBI (*Neher et al., 2014*) and mutational load (*Luksza and Lässig, 2014*). For simulated A/H3N2-like populations, we produced forecasts with the "true fitness" model that relies on the normalized fitness value of each simulated sample.

Each forecast generated a predicted future frequency per sequence in the initial timepoint's tree. As in our prior work, we calculated the earth mover's distance (*Rubner et al., 1998*) between the predicted and observed future populations using HA AA sequences from initial and future timepoints, predicted future frequencies from the initial timepoint, and observed future frequencies from future timepoint. For the future timepoint, we used data from the "no lag" scenario as our truth set, regardless of the lag scenario for the initial timepoint. This design allowed us to measure the effect of ideal and realistic submission lags on forecast accuracy relative to a scenario with no lags.

## Defining clades

Official clade definitions do not exist for all time periods of our analysis of A/H3N2 populations and do not exist at all for simulated A/H3N2-like populations. Therefore, we defined clades de novo for both population types with the same clade assignment algorithm used to produce "subclades" for recent seasonal influenza vaccine composition meeting reports (*Huddleston et al., 2024*). The complete algorithm description and implementation is available at https://github.com/neherlab/flu_clades (*Neher, 2023*). Briefly, the algorithm scores each node in a phylogenetic tree based on three criteria including the number of child nodes descending from the current node, the number of epitope substitutions on the branch leading to the current node, and the number of AA mutations since the last clade assigned to an ancestor in the tree. After assigning and normalizing scores, the algorithm traverses the tree in preorder, assigning clade labels to each internal node whose score exceeds a predefined threshold of 1.0. Clade labels follow a hierarchical nomenclature inspired by Pangolin (*O'Toole et al., 2021*) such that the first clade in the tree is named "A" and its first immediate descendant is named "A.1". For each population type, we applied this algorithm to a single phylogeny representing all HA sequences present in our analysis. This approach allowed us to produce a single clade assignment per sequence and easily identify related sequences between initial and future timepoints using the hierarchical clade nomenclature.

## Estimating current and future clade frequencies

We estimated clade frequencies with a kernel density estimation (KDE) approach as previously described (*Huddleston et al., 2020*) with the `augur frequencies` command (*Huddleston et al., 2021*). Briefly, we represented each sequence in a given phylogeny by a Gaussian kernel with a mean at the sequence's collection date and a variance of 2 months. We estimated the frequency of each sequence at each timepoint by calculating the probability density function of each KDE at that timepoint and normalizing the resulting values to sum to one.

We calculated clade frequencies for each initial timepoint in our analysis by first summing the frequencies of individual sequences in a given timepoint's tree by the clade assigned to each sequence and then summing the frequencies for each clade and its descendants to obtain nested clade frequencies. To inspect the effects of submission lags on clade frequency estimates, we calculated the clade frequency error per timepoint and clade by subtracting the clade frequency estimated with ideal or realistic lagged sequence submission from the corresponding clade frequency without lags. We compared the effects of submission lags for clades of different sizes by filtering clades by their frequency estimated without lags to small clades (>0% and <10%) and large clades (≥10%).

To estimate the accuracy of clade frequency forecasts, we needed to calculate the predicted and observed future clade frequencies for each combination of lag type, initial timepoint, and future timepoint in the analysis. We calculated predicted future frequencies for all clades that existed at given initial timepoint and lag types by first summing the predicted future frequency per sequence by the clade assigned to each sequence and then summing the predicted frequencies for each clade and its descendants. Clades that existed at any given future timepoint were not always represented at a corresponding initial timepoint either because the clades had not emerged yet or sequences for those clades had a lagged submission. For this reason, we calculated observed future clade frequencies in a multi-step process. First, we calculated the frequencies of clades observed at the future timepoint without submission lag by summing the individual frequencies of all sequences in each clade. Then, we mapped each future clade to its most derived ancestral clade that circulated at the initial timepoint by progressively removing suffixes from the future clade's label until we found a match in the initial timepoint. For example, if the future timepoint had a clade named A.1.1.3 and the initial timepoint had the ancestral clade A.1, we would test for the presence of A.1.1.3, A.1.1, and A.1 at the initial timepoint until we found a match. The hierarchical nature of the clade assignment algorithm guaranteed that each future clade mapped directly to a clade at each initial timepoint and lag type. Finally, we summed the frequencies of future clades by their corresponding initial clades to get the observed future frequencies of clades circulating at the initial timepoint. We calculated the accuracy of clade frequency forecasts as the difference between the predicted and observed future clade frequencies.

## Acknowledgements

We gratefully acknowledge the authors, originating and submitting laboratories of the sequences from the GISAID EpiFlu Database (*Shu and McCauley, 2017*) on which this research is based. A list of sequence accessions, authors, and labs appears in the Supplemental Material. We thank Katie Kistler and Marlin Figgins for their comments on early versions of this manuscript and Richard A Neher for the development of tools for hierarchical clade nomenclature. This work was funded by NIAID R01 AI165821-01. TB is a Howard Hughes Medical Institute Investigator.

## Additional information

### Funding

| Funder | Grant reference number | Author |
|---|---|---|
| National Institute of Allergy and Infectious Diseases | R01 AI165821-01 | Trevor Bedford |
| Howard Hughes Medical Institute | | Trevor Bedford |

The funders had no role in study design, data collection and interpretation, or the decision to submit the work for publication.

### Author contributions

John Huddleston, Conceptualization, Resources, Data curation, Software, Formal analysis, Funding acquisition, Validation, Investigation, Visualization, Methodology, Writing – original draft, Project administration, Writing – review and editing; Trevor Bedford, Supervision, Funding acquisition, Writing – review and editing

## Author ORCIDs

John Huddleston https://orcid.org/0000-0002-4250-2063
Trevor Bedford https://orcid.org/0000-0002-4039-5794

Reviewer #1 (Public review): https://doi.org/10.7554/eLife.104282.3.sa1
Author response https://doi.org/10.7554/eLife.104282.3.sa2

## Additional files

### Supplementary files

MDAR checklist

Supplementary file 1. GISAID accessions and metadata including originating and submitting labs for natural strains used across all timepoints.

### Data availability

Sequence data are available from the GISAID EpiFlu Database using accessions provided in *Supplementary file 1*. Source code for the analysis workflow and manuscript are available in the project's GitHub repository (https://github.com/blab/flu-forecasting-delays, copy archived at *Huddleston, 2025*). Supplemental data are available on Zenodo at https://doi.org/10.5281/zenodo.17259448.

The following dataset was generated:

| Author(s) | Year | Dataset title | Dataset URL | Database and Identifier |
|---|---|---|---|---|
| Huddleston J, Bedford T | 2025 | Supplementary data for "Timely vaccine strain selection and genomic surveillance improves evolutionary forecast accuracy of seasonal influenza A/H3N2" | https://doi.org/10.5281/zenodo.17259448 | Zenodo, 10.5281/zenodo.17259448 |

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
