## [Editor Report · eLife Assessment]

This study investigates the influence of genomic information and timing of vaccine strain selection on the accuracy of influenza A/H3N2 forecasting. The authors utilized appropriate statistical methods and have provided **convincing** evidence, which amounts to an **important** contribution to the evidence base. Substantial revisions have been made to the manuscript and issues of concern have been clarified, with the necessary study limitations appropriately discussed.

---

## [Referee Report · Reviewer #1 (Public review)]

Summary:

In the paper, the authors investigate how the availability of genomic information and the timing of vaccine strain selection influence the accuracy of influenza A/H3N2 forecasting. The manuscript presents three key findings:

(1) Using real and simulated data, the authors demonstrate that shortening the forecasting horizon and reducing submission delays for sharing genomic data improve the accuracy of virus forecasting.

(2) Reducing submission delays also enhances estimates of current clade frequencies.

(3) Shorter forecasting horizons, for example allowed by the proposed use of "faster" vaccine platforms such as mRNA, result in the most significant improvements in forecasting accuracy.

Strengths:

The authors present a robust analysis, using statistical methods based on previously published genetic based techniques to forecast influenza evolution. Optimizing prediction methods is crucial from both scientific and public health perspectives. The use of simulated as well as real genetic data (collected between April 1, 2005, and October 1, 2019) to assess the effects of shorter forecasting horizons and reduced submission delays is valuable and provides a comprehensive dataset. Moreover, the accompanying code is openly available on GitHub and is well-documented.

Limitations of the authors genomic-data-only approach are discussed in depth and within the context of existing literature. In particular, the impact of subsampling, necessary for computational reasons in this study, or restriction to Northen/Southern Hemisphere data is explored and discussed.

Weaknesses:

Although the authors acknowledge these limitations in their discussion, the impact of the analysis is somewhat constrained by its exclusive reliance on methods using genomic information, without incorporating or testing the impact of phenotypic data. The analysis with respect to more integrative models remains open and the authors do not empirically validate how the inclusion of phenotypic information might alter or impact the findings. Instead, we must rely on the authors' expectation that their findings are expected to hold across different forecasting models, including those integrating both phenotypic and genetic data. This expectation, while reasonable, remains untested within the scope of the current study.

Comments on latest version:

Thanks to the authors for the revised version of the manuscript, which addresses and clarifies all of my previously raised points.

In particular, the exploration of how subsampling of genomic information, hemisphere-specific forecasting, and the check for time dependence potentially influence the findings is now included and adds to the discussion. The manuscript also benefits from a look at these limitations when relying only on genomic data.

The authors have carefully placed these limitations within the context of existing literature, especially on the raised concern to not include phenotypic data. As a minor comment, the conclusion that the findings potentially stay across different forecasting models, including those integrating both phenotypic and genetic data, rely on the author's expectation. While this expectation might be plausible, it remains to be validated empirically in future work.

---

## [Author Response]

The following is the authors’ response to the original reviews.

**Reviewer #1 (Public review)**
Summary:In the paper, the authors investigate how the availability of genomic information and the timing of vaccine strain selection influence the accuracy of influenza A/H3N2 forecasting. The manuscript presents three key findings:(1) Using real and simulated data, the authors demonstrate that shortening the forecasting horizon and reducing submission delays for sharing genomic data improve the accuracy of virus forecasting.(2) Reducing submission delays also enhances estimates of current clade frequencies.(3) Shorter forecasting horizons, for example, allowed by the proposed use of "faster" vaccine platforms such as mRNA, resulting in the most significant improvements in forecasting accuracy.Strengths:The authors present a robust analysis, using statistical methods based on previously published genetic-based techniques to forecast influenza evolution. Optimizing prediction methods is crucial from both scientific and public health perspectives. The use of simulated as well as real genetic data (collected between April 1, 2005, and October 1, 2019) to assess the effects of shorter forecasting horizons and reduced submission delays is valuable and provides a comprehensive dataset. Moreover, the accompanying code is openly available on GitHub and is well-documented.

Thank you for this summary! We worked hard to make this analysis robust, reproducible, and open source.

Weaknesses:While the study addresses a critical public health issue related to vaccine strain selection and explores potential improvements, its impact is somewhat constrained by its exclusive reliance on predictive methods using genomic information, without incorporating phenotypic data. The analysis remains at a high level, lacking a detailed exploration of factors such as the genetic distance of antigenic sites.

We are glad to see this acknowledgment of the critical public health issue we've addressed in this project. The goal for this study was to test effects of counterfactual scenarios with realistic public health interventions and not to introduce methodological improvements to forecasting methods. The final forecasting model we analyzed in this study (lines 301-330 and Figure 6) was effectively an "oracle" model that produced the optimal forecast for each given current and future timepoint. We expect any methodological improvements to forecasting models to converge toward the patterns we observed in this final section of the results.

We've addressed the reviewer's concerns in more detail in response to their numbered comments 4 and 5 below.

Another limitation is the subsampling of the available dataset, which reduces several tens of thousands of sequences to just 90 sequences per month with even sampling across regions. This approach, possibly due to computational constraints, might overlook potential effects of regional biases in clade distribution that could be significant. The effect of dataset sampling on presented findings remains unexplored. Although the authors acknowledge limitations in their discussion section, the depth of the analysis could be improved to provide a more comprehensive understanding of the underlying dynamics and their effects.

We have addressed this comment in the numbered comment 1 below.

Suggestions to enhance the depth of the manuscript:

Thank you again for these thoughtful suggestions. They have encouraged us to revisit aspects of this project that we had overlooked by being too close to it and have helped us improve the paper's quality.

(1) Subsampling and Sampling Strategies: It would be valuable to comment on the rationale behind the strong subsampling of the available GISAID data. A discussion of the potential effects of different sampling strategies is necessary. Additionally, assessing the stability of the results under alternative sequence sampling strategies would strengthen the robustness of the conclusions.

We agree with the reviewer's point that our subsampled sequences only represent a fraction of those available in the GISAID EpiFlu database and that a more complete representation would be ideal. We designed the subsampling approach we used in this study for two primary reasons.

(1) First, we sought to minimize known regional and temporal biases in sequence availability. For example, North America and Europe are strongly overrepresented in the GISAID EpiFlu database, while Africa and Asia are underrepresented (Figure 1A). Additionally, the number of sequences in the database has increased every year since 2010, causing later years in this study period to be overrepresented compared to earlier years. A major limitation of our original forecasting model from Huddleston et al. 2020 is its inability to explicitly estimate geographic-specific clade fitnesses. Because of this limitation, we trained that original model on evenly subsampled sequences across space and time. We used the same approach in this study to allow us to reuse that previously trained forecasting model. Despite this strong subsampling approach, we still selected an average of 50% of all available sequences across all 10 regions and the entire study period (Figure 1B). Europe and North America were most strongly downsampled with only 7% and 8% of their total sequences selected for the study, respectively. In contrast, we selected 91% of all sequences from Southeast Asia.

(2) Second, our forecasting model relies on the inference of time-scaled phylogenetic trees which are computationally intensive to infer. While new methods like CMAPLE (Ly-Trong et al. 2024) would allow us to rapidly infer divergence trees, methods to infer time trees still do not scale well to more than ~20,000 samples. The subsampling approach we used in this study allowed us to build the 35 six-year H3N2 HA trees we needed to test our forecasting model in a reasonable amount of time.

We have expanded our description of this rationale for our subsampling approach in the discussion and described the potential effects of geographic and temporal biases on forecasting model predictions (lines 360-376). Our original discussion read:

"Another immediate improvement would be to develop models that can use all available data in a way that properly accounts for geographic and temporal biases. Current models based on phylogenetic trees need to evenly sample the diversity of currently circulating viruses to produce unbiased trees in a reasonable amount of time. Models that could estimate sample fitness and compare predicted and future populations without trees could use more available sequence data and reduce the uncertainty in current and future clade frequencies."

The section now reads:

"Another immediate improvement would be to develop models that can use all available data in a way that properly accounts for geographic and temporal biases. For example, virus samples from North America and Europe are overrepresented in the GISAID EpiFlu database, while samples from Africa and Asia are underrepresented (McCarron et al. 2022). As new H3N2 epidemics often originate from East and Southeast Asia and burn out in North America and Europe (Bedford et al. 2015), models that do not account for this geographic bias are more likely to incorrectly predict the success of lower fitness variants circulating in overrepresented regions and miss higher fitness variants emerging from underrepresented regions. Additionally, the number of H3N2 HA sequences per year in the GISAID EpiFlu database has increased consistently since 2010, creating a temporal bias where any given season a model forecasts to will have more sequences available than the season from which forecasts occur. The model we used in this study does not explicitly account for geographic variability of viral fitness and relies on time-scaled phylogenetic trees which can be computationally costly to infer for large sample sizes. As a result, we needed to evenly sample the diversity of currently circulating viruses to produce unbiased trees in a reasonable amount of time. Models that could estimate viral fitness per geographic region without inferring trees could use more available sequence data and reduce the uncertainty in current and future clade frequencies."

We also added a brief explanation of our subsampling method to the corresponding section of the methods (lines 411-415). These lines read:

"This sampling approach accounts for known regional biases in sequence availability through time (McCarron et al. 2022) and makes inference of divergence and time trees computationally tractable. This approach also exactly matches our previous study where we first trained the forecast models used in this study (Huddleston et al. 2020), allowing us to reuse those previously trained models."

Although our forecast model is limited to a small proportion of sequences that we evenly sample across regions and time, we agree that we could improve the robustness of our conclusions by repeating our analysis for different subsets of the available data. To assess the stability of the results under alternative sequence sampling strategies, we ran a second replicate of our entire analysis of natural H3N2 populations with three times as many sequences per month (270) than our original replicate. With this approach, we selected between 17% (Europe) and 97% (Southeast Asia) of all sequences per region with an average of 72% and median of 83% (Figure 1C). We compared the effects of realistic interventions for this high-density subsampling analysis with the effects from the original subsampling analysis (Figure 6). We have added the results from this analysis to the main text (lines 313-321) which now reads:

"For natural A/H3N2 populations, the average improvement of the vaccine intervention was 1.1 AAs and the improvement of the surveillance intervention was 0.27 AAs or approximately 25% of the vaccine intervention. The average improvement of both interventions was only slightly less than additive at 1.28 AAs. To verify the robustness of these results, we replicated our entire analysis of A/H3N2 populations using a subsampling scheme that tripled the number of viruses selected per month from 90 to 270 (Figure 1—figure supplement 4C). We found the same pattern with this replication analysis, with average improvements of 0.93 AAs for the vaccine intervention, 0.21 AAs for the surveillance intervention, and 1.14 AAs for both interventions (Figure 6—figure supplement 2)."

We updated our revised manuscript to include the summary of sequences available and subsampled as Figure 1—figure supplement 4 and the effects of interventions with the high-density analysis as Figure 6—figure supplement 2. For reference, we have included Figure 2 showing both the original Figure 6 (original subsampling) and Figure 6—figure supplement 2 (high-density subsampling).

(2) Time-Dependent Effects: Are there time-dependent patterns in the findings? For example, do the effects of submission lag or forecasting horizon differ across time periods, such as [2005-2010, +2010-2015,2015-2018]? This analysis could be particularly interesting given the emergence of co-circulation of clades 3c.2 and 3c.3 around 2012, which marked a shift to less "linear" evolutionary patterns over many years in influenza A/H3N2.

This is an interesting question that we overlooked by focusing on the broader trends in the predictability of A/H3N2 evolution. The effects of realistic interventions that we report in Figure 6 span future timepoints of 2012-04-01 to 2019-10-01. Since H1N1pdm emerged in 2009 and 3c3 started cocirculating with 3c2 in 2012, we can't inspect effects for the specific epochs mentioned above. However, there have been many periods during this time span where the number of cocirculating clades varied in ways that could affect forecast accuracy. The streamgraph, Author response image 1, shows the variation in clade frequencies from the "full tree" that we used to define clades for A/H3N2 populations.

**Author response image 1. sa2fig1:** Streamgraph of clade frequencies for A/H3N2 populations demonstrating variability of clade cocirculation through time.

We might expect that forecasting models would struggle to accurately predict future timepoints with higher clade diversity, since much of that diversity would not have existed at the time of the forecast. We might also expect faster surveillance to improve our ability to detect that future variation by detecting those variants at low frequency instead of missing them completely.

To test this hypothesis, we calculated the Shannon entropy of clade frequencies per future timepoint represented in Figure 6 (under no submission lag) and plotted the change in optimal distance to the predicted future by the entropy per timepoint. If there was an effect of future clade complexity on forecast accuracy, we expected greater improvements from interventions to be associated with higher future entropy.

There was a trend for some of the greatest improvements per intervention to occur at higher future clade entropy timepoints, but we didn’t find a strong relationship between clade entropy and improvement in forecast accuracy by any intervention (Figure 4). The highest correlation was for improved surveillance (Pearson r=0.24).

We have added this figure to the revised manuscript as Figure 6—figure supplement 3 and updated the results (lines 321-323) to reflect the patterns we described above. The updated results (which partially includes our response to the next reviewer comment) read:

"These effects of realistic interventions appeared consistent across the range of genetic diversity at future timepoints (Figure 6—figure supplement 3) and for future seasons occurring in both Northern and Southern Hemispheres (Figure 6—figure supplement 4)."

(3) Hemisphere-Specific Forecasting: Do submission lags or forecasting horizons show different performance when predicting Northern versus Southern Hemisphere viral populations? Exploring this distinction could add significant value to the analysis, given the seasonal differences in influenza circulation.

Similar to the question above, we can replot the improvements in optimal distances to the future for the realistic interventions, grouping values by the hemisphere that has an active season in each future timepoint. Much like we expected forecasts to be less accurate when predicting into a highly diverse season, we might also expect forecasts to be less accurate when predicting into a season for a more densely populated hemisphere. Specifically, we expected that realistic interventions would improve forecast accuracy more for Northern Hemisphere seasons than Southern Hemisphere seasons. For this analysis, we labeled future timepoints that occurred in October or January as "Northern" and those that occurred in April or July as "Southern". We plotted effects of interventions on optimal distances to the future by intervention and hemisphere.

In contrast to our original expectation, we found a slightly higher median improvement for the Southern Hemisphere seasons under both of the interventions that improved the vaccine timeline (Figure 5). The median improvement for the combined intervention was 1.42 AAs in the Southern Hemisphere and 0.93 AAs in the Northern Hemisphere. Similarly, the improvement with the "improved vaccine" intervention was 1.03 AAs in the South and 0.74 AAs in the North. However, the range of improvements per intervention was greater for the Northern Hemisphere across all interventions. The median increase in forecast accuracy was similar for both hemispheres in the improved surveillance intervention, with a single Northern Hemisphere season showing an unusually greater improvement that was also associated with higher clade entropy (Figure 4). These results suggest that both an improved vaccine development timeline and more timely sequence submissions would most improve forecast accuracy for Southern Hemisphere seasons compared to Northern Hemisphere seasons.

We have added this figure to the revised manuscript as Figure 6—figure supplement 4 and updated the results (lines 321-326) to reflect the patterns we described above. The new lines in the results read:

"These effects of realistic interventions appeared consistent across the range of genetic diversity at future timepoints (Figure 6—figure supplement 3) and for future seasons occurring in both Northern and Southern Hemispheres (Figure 6—figure supplement 4). We noted a slightly greater median improvement in forecast accuracy associated with both improved vaccine interventions for the Southern Hemisphere seasons (1.03 and 1.42 AAs) compared to the Northern Hemisphere seasons (0.74 and 0.93 AAs)."

(4) Antigenic Sites and Submission Delays: It would be interesting to investigate whether incorporating antigenic site information in the distance metric amplifies or diminishes the observed effects of submission delays. Such an analysis could provide a first glance at how antigenic evolution interacts with forecasting timelines.

This would be an interesting area to explore. One hypothesis along these lines would be that if (1) viruses with more substitutions at antigenic sites are more likely to represent the future population and (2) viruses with more antigenic substitutions originate in specific geographic locations and (3) submissions of sequences for those viruses are more likely to be lagged due to their geographic origin, then (4) decreasing submission lags should improve our forecasting accuracy by detecting antigenically-important sequences earlier. If there is not a direct link between viruses that are more likely to represent the future and higher submission lags, we would not expect to see any additional effect of reducing submission lags for antigenic sites. Based on our work in Huddleston et al. 2020, it is also not clear that assumption 1 above is consistently true, since the specific antigenic sites associated with high fitness change over time. In that earlier work, we found that models based on these antigenic (or "epitope") sites could only accurately predict the future when the relevant sites for viral success were known in advance. This result was shown by our "oracle" model which accurately predicted the future during the model validation period when it knew which sites were associated with success and failed to predict the future in the test period when the relevant sites for success had changed (Figure 6).

To test the hypothesis above, we would need sequences to have submission lags that reflect their geographic origin. For this current study, we intentionally decoupled submission lags from geographic origin to allow inclusion of historical A/H3N2 HA sequences that were originally submitted as part of scientific publications and not as part of modern routine surveillance. As a result, the original submission dates for many sequences are unrealistically lagged compared to surveillance sequences.

(5) Incorporation of Phenotypic Data: The authors should provide a rationale for their choice of a genetic-information-only approach, rather than a model that integrates phenotypic data. Previous studies, such as Huddleston et al. (2020, eLife), demonstrate that models combining genetic and phenotypic data improve forecasts of seasonal influenza A/H3N2 evolution. It would be interesting to probe the here observed effects in a more recent model.

The primary goal of this study was not to test methodological improvements to forecasting models but to test the effects of realistic public health policy changes that could alter forecast horizons and sequence availability. Most influenza collaborating centers use a "sequence-first" approach where they sequence viral isolates first and use those sequences to prioritize viruses for phenotypic characterization (Hampson et al. 2017). The additional lag in availability of phenotypic data means that a forecasting model based on genetic and phenotypic data will necessarily have a greater lag in data availability than a model based on genetic data only. Since the policy changes we're testing in this study only affect the availability of sequence data and not phenotypic data, we chose to test the relative effects of policy changes on sequence-based forecasting models.

We have updated the abstract (lines 18-26 and 30-32), introduction (lines 87-88), and discussion (lines 332-334) to emphasize the focus of this study on effects of policy changes. The updated abstract lines read as follows with new content in bold:

"Despite continued methodological improvements to long-term forecasting models, these constraints of a 12-month forecast horizon and 3-month average submission lags impose an upper bound on any model's accuracy. The global response to the SARS-CoV-2 pandemic revealed that the adoption of modern vaccine technology like mRNA vaccines can reduce how far we need to forecast into the future to 6 months or less and that expanded support for sequencing can reduce submission lags to GISAID to 1 month on average. To determine whether these public health policy changes could improve long-term forecasts for seasonal influenza, we quantified the effects of reducing forecast horizons and submission lags on the accuracy of forecasts for A/H3N2 populations. We found that reducing forecast horizons from 12 months to 6 or 3 months reduced average absolute forecasting errors to 25% and 50% of the 12-month average, respectively. Reducing submission lags provided little improvement to forecasting accuracy but decreased the uncertainty in current clade frequencies by 50%. These results show the potential to substantially improve the accuracy of existing influenza forecasting models through the public health policy changes of modernizing influenza vaccine development and increasing global sequencing capacity."

The updated introduction now reads:

"These technological and public health policy changes in response to SARS-CoV-2 suggest that we could realistically expect the same outcomes for seasonal influenza."

The updated discussion now reads:

"In this work, we showed that realistic public health policy changes that decrease the time to develop new vaccines for seasonal influenza A/H3N2 and decrease submission lags of HA sequences to public databases could improve our estimates of future and current populations, respectively."

We have also updated the introduction (lines 57-65) and the discussion (lines 345-348) to specifically address the use of sequence-based models instead of sequence-and-phenotype models. The updated introduction now reads:

"For this reason, the decision process is partially informed by computational models that attempt to predict the genetic composition of seasonal influenza populations 12 months in the future (Morris et al. 2018). The earliest of these models predicted future influenza populations from HA sequences alone (Luksza and Lassig 2014, Neher et al. 2014, Steinbruck et al. 2014). Recent models include phenotypic data from serological experiments (Morris et al. 2018, Huddleston et al. 2020, Meijers et al. 2023, Meijers et al. 2025). Since most serological experiments occur after genetic sequencing (Hampson et al. 2017) and all forecasting models depend on HA sequences to determine the viruses circulating at the time of a forecast, sequence availability is the initial limiting factor for any influenza forecasts."

The updated discussion now reads:

"Since all models to date rely on currently available HA sequences to determine the clades to be forecasted, we expect that decreasing forecast horizons and submission lags will have similar relative effect sizes across all forecasting models including those that integrate phenotypic and genetic data."

**Reviewer #2 (Public review):**
Summary:The authors have examined the effects of two parameters that could improve their clade forecasting predictions for A(H3N2) seasonal influenza viruses based solely on analysis of haemagglutinin gene sequences deposited on the GISAID Epiflu database. Sequences were analysed from viruses collected between April 1, 2005 and October 1, 2019. The parameters they investigated were various lag periods (0, 1, 3 months) for sequences to be deposited in GISAID from the time the viruses were sequenced. The second parameter was the time the forecast was accurate over projecting forward (for 3,6,9,12 months). Their conclusion (not surprisingly) was that "the single most valuable intervention we could make to improve forecast accuracy would be to reduce the forecast horizon to 6 months or less through more rapid vaccine development". This is not practical using conventional influenza vaccine production and regulatory procedures. Nevertheless, this study does identify some practical steps that could improve the accuracy and utility of forecasting such as a few suggested modifications by the authors such as "..... changing the start and end times of our long-term forecasts. We could change our forecasting target from the middle of the next season to the beginning of the season, reducing the forecast horizon from 12 to 9 months.'Strengths:The authors are very familiar with the type of forecasting tools used in this analysis (LBI and mutational load models) and the processes used currently for influenza vaccine virus selection by the WHO committees having participated in a number of WHO Influenza Vaccine Consultation meetings for both the Southern and Northern Hemispheres.Weaknesses:The conclusion of limiting the forecasting to 6 months would only be achievable from the current influenza vaccine production platforms with mRNA. However, there are no currently approved mRNA influenza vaccines, and mRNA influenza vaccines have also yet to demonstrate their real-world efficacy, longevity, and cost-effectiveness and therefore are only a potential platform for a future influenza vaccine. Hence other avenues to improve the forecasting should be investigated.

We recognize that there are no approved mRNA influenza vaccines right now. However, multiple mRNA vaccines have completed phase 3 trials indicating that these vaccines could realistically become available in the next few years. A primary goal of our study was to quantify the effects of switching to a vaccine platform with a shorter timeline than the status quo. Our results should further motivate the adoption of any modern vaccine platform that can produce safe and effective vaccines more quickly than the egg-passaged standard. We have updated the introduction (lines 88-91) to note the mRNA vaccines that have completed phase 3 trials. The new sentence in the introduction reads:

"Work on mRNA vaccines for influenza viruses dates back over a decade (Petsch et al. 2012, Brazzoli et al. 2016, Pardi et al. 2018, Feldman et al. 2019), and multiple vaccines have completed phase 3 trials by early 2025 (Soens et al. 2025, Pfizer 2022)."

While it is inevitable that more influenza HA sequences will become available over time a better understanding of where new influenza variants emerge would enable a higher weighting to be used for those countries rather than giving an equal weighting to all HA sequences.

This is definitely an important point to consider. The best estimates to date (Russell et al. 2008, Bedford et al. 2015) suggest that most successful variants emerge from East or Southeast Asia. In contrast, most available HA sequence data comes from Europe and North America (Figure 1A). Our subsampling method explicitly tries to address this regional bias in data availability by evenly sampling sequences from 10 different regions including four distinct East Asian regions (China, Japan/Korea, South Asia, and Southeast Asia). Instead of weighting all HA sequences equally, this sampling approach ensures that HA sequences from important distinct regions appear in our analysis.

We have updated our methods (lines 411-423) to better describe the motivation of our subsampling approach and proportions of regions sampled with our original approach (90 viruses per month) and a second high-density sampling approach (270 viruses per month). These new lines read:

"This sampling approach accounts for known regional biases in sequence availability through time (McCarron et al. 2022) and makes inference of divergence and time trees computationally tractable. This approach also exactly matches our previous study where we first trained the forecast models used in this study (Huddleston et al. 2020), allowing us to reuse those previously trained models. With this subsampling approach, we selected between 7% (Europe) and 91% (Southeast Asia) of all available sequences per region across the entire study period with an average of 50% and median of 52% across all 10 regions (Figure 1—figure Supplement 4). To verify the reproducibility and robustness of our results, we reran the full forecasting analysis with a high-density subsampling scheme that selected 270 sequences per month with the same even sampling across regions and time as the original scheme. With this approach, we selected between 17% (Europe) and 97% (Southeast Asia) of all available sequences per region with an average of 72% sampled and a median of 83% (Figure 1—figure Supplement 4C)."

We added Figure 1—figure Supplement 4 to document the regional biases in sequence availability and the proportions of sequences we selected per region and year.

Also, other groups are considering neuraminidase sequences and how these contribute to the emergence of new or potentially predominant clades.

We agree that accounting for antigenic evolution of neuraminidase is a promising path to improving forecasting models. We chose to focus on hemagglutinin sequences for several reasons, though. First, hemagglutinin is the only protein whose content is standardized in the influenza vaccine (Yamayoshi and Kawaoka 2019), so vaccine strain selection does not account for a specific neuraminidase. Additionally, as we noted in response to Reviewer 1 above, the goal of this study was to test effects of counterfactual scenarios with realistic public health interventions and not to introduce methodological improvements to forecasting models like the inclusion of neuraminidase sequences.

We have updated the introduction to provide the additional context about hemagglutinin's outsized role in the current vaccine development process (lines 40-44):

"The dominant influenza vaccine platform is an inactivated whole virus vaccine grown in chicken eggs (Wong and Webby, 2013) which takes 6 to 8 months to develop, contains a single representative vaccine virus per seasonal influenza subtype including A/H1N1pdm, A/H3N2, and B/Victoria (Morris et al., 2018), and for which only the HA protein content is standardized (Yamayoshi and Kawaoka, 2019)."

We have updated the abstract (lines 18-26 and 30-32), introduction (lines 87-88), and discussion (lines 332-334) to emphasize our goal of testing effects of public health policy changes on forecasting accuracy rather than methodological changes. The updated abstract lines read as follows with new content in bold:

"Despite continued methodological improvements to long-term forecasting models, these constraints of a 12-month forecast horizon and 3-month average submission lags impose an upper bound on any model's accuracy. The global response to the SARS-CoV-2 pandemic revealed that the adoption of modern vaccine technology like mRNA vaccines can reduce how far we need to forecast into the future to 6 months or less and that expanded support for sequencing can reduce submission lags to GISAID to 1 month on average. To determine whether these public health policy changes could improve long-term forecasts for seasonal influenza, we quantified the effects of reducing forecast horizons and submission lags on the accuracy of forecasts for A/H3N2 populations. We found that reducing forecast horizons from 12 months to 6 or 3 months reduced average absolute forecasting errors to 25% and 50% of the 12-month average, respectively. Reducing submission lags provided little improvement to forecasting accuracy but decreased the uncertainty in current clade frequencies by 50%. These results show the potential to substantially improve the accuracy of existing influenza forecasting models through the public health policy changes of modernizing influenza vaccine development and increasing global sequencing capacity."

The updated introduction now reads:

"These technological and public health policy changes in response to SARS-CoV-2 suggest that we could realistically expect the same outcomes for seasonal influenza."

The updated discussion now reads:

"In this work, we showed that realistic public health policy changes that decrease the time to develop new vaccines for seasonal influenza A/H3N2 and decrease submission lags of HA sequences to public databases could improve our estimates of future and current populations, respectively."

Figure 1a. I don't understand why the orange dot 1-month lag appears to be on the same scale as the 3-month/ideal timeline.

We apologize for the confusion with this figure. Our original goal was to show how the two factors in our study design (forecast horizons and sequence submission lags) interact with each other by showing an example of 3-month forecasts made with no lag (blue), ideal lag (orange), and realistic lag (green). To clarify these two factors, we have removed the two lines at the 3-month forecast horizon for the ideal and realistic lags and have updated the caption to reflect this simplification. The new figure looks like this:

The authors should expand on the line "The finding of even a few sequences with a potentially important antigenic substitution could be enough to inform choices of vaccine candidate viruses." While people familiar with the VCM process will understand the implications of this statement the average reader will not fully understand the implications of this statement. Not only will it inform but it will allow the early production of vaccine seeds and reassortants that can be used in conventional vaccine production platforms if these early predictions were consolidated by the time of the VCM. This is because of the time it takes to isolate viruses, make reassortants and test them - usually a month or more is needed at a minimum.

Thank you for pointing out this unclear section of the discussion. We have rewritten this section, dropping the mention of prospective measurements of antigenic escape which now feels off-topic and moving the point about early detection of important antigenic substitutions to immediately follow the description of the candidate vaccine development timeline. This new placement should clarify the direct causal relationship between early detection and better choices of vaccine candidates. The original discussion section read:

"For example, virologists must choose potential vaccine candidates from the diversity of circulating clades well in advance of vaccine composition meetings to have time to grow virus in cells and eggs and measure antigenic drift with serological assays (Morris et al., 2018; Loes et al., 2024). Similarly, prospective measurements of antigenic escape from human sera allow researchers to predict substitutions that could escape global immunity (Lee et al., 2019; Greaney et al., 2022; Welsh et al., 2023). The finding of even a few sequences with a potentially important antigenic substitution could be enough to inform choices of vaccine candidate viruses."

The new section (lines 386-391) now reads:

"For example, virologists must choose potential vaccine candidates from the diversity of circulating clades months in advance of vaccine composition meetings to have time to grow virus in cells and eggs and measure antigenic drift with serological assays (Morris et al. 2018; Loes et al. 2024). Earlier detection of viral sequences with important antigenic substitutions could determine whether corresponding vaccine candidates are available at the time of the vaccine selection meeting or not."

A few lines in the discussion on current approaches being used to add to just the HA sequence analysis of H3N2 viruses (ferret/human sera reactivity) would be welcome.

We have added the following sentences to the last paragraph (lines 391-397) to note recent methodological advances in estimating influenza fitness and the relationship these advances have to timely genomic surveillance.

"Newer methods to estimate influenza fitness use experimental measurements of viral escape from human sera (Lee et al., 2019; Welsh et al., 2024; Meijers et al., 2025; Kikawa et al., 2025), measurements of viral stability and cell entry (Yu et al., 2025), or sequences from neuraminidase, the other primary surface protein associated with antigenic drift (Meijers et al., 2025). These methodological improvements all depend fundamentally on timely genomic surveillance efforts and the GISAID EpiFlu database to identify relevant influenza variants to include in their experiments."